# Transpiration rates from a̶ mature *Eucalyptus grandis* x *E. nitens* clonal hybrid and *Pinus elliottii* plantations near the Two Streams Research Catchment, South Africa

Nkosinathi D. Kaptein[1], Colin S. Everson[2,4], Alistair D. Clulow[1,2], Michele L. Toucher[2,3], Ilaria Germishuizen[5]

[1]Discipline of Agrometeorology, University of KwaZulu-Natal, Pietermaritzburg, 3209, South Africa
[2]Centre for Water Resources Research, University of KwaZulu-Natal, Pietermaritzburg, 3209, South Africa
[3]Grasslands-Forests-Wetlands Node, South African Environmental Observation Network, Pietermaritzburg, 3201, South Africa
[4]Department Plant and Soil Sciences, University of Pretoria, Pretoria, South Africa
[5]Institute for Commercial Forestry Research, Scottsville, 3201, South Africa

*Correspondence to*: Nkosinathi D. Kaptein (kapteinnd@gmail.com)

**Abstract.** Pine plantations are the dominant species currently planted within the South African commercial forestry industry. Improvements in bioeconomy markets for dissolving wood pulp products have seen an expansion in fast-growing *Eucalyptus* plantations due to their higher productivity rates and better pulping properties than pine. This has raised concerns regarding the expansion of *Eucalyptus* plantations and how they will affect water resources as they have been reported to have higher water-use (quantified using transpiration rates) than pine. We measured transpiration rates (mm year$^{-1}$), diameter at breast height (quantified as quadratic mean diameter, $D_q$, m) and leaf area index of an eight-year-old *Eucalyptus grandis* x *Eucalyptus nitens* clonal hybrid (*GN*) and a twenty-year-old *Pinus elliottii*. Transpiration rates were measured for two consecutive hydrological years (2019/ 20 and 2020/ 21) using a heat ratio sap-flow method, calibrated against a lysimeter. In the 2019/ 20 year, annual transpiration for *P. elliottii* exceeded *GN* by 28%, while for the 2020/ 21 hydrological year, there was no significant difference between the transpiration of the two species, despite a 17 and 21% greater leaf area index for *P. elliottii* than *GN* in 2019/ 20 and 2020/ 21 measurement years, respectively. Quadratic mean diameter increments were statistically similar ($p > 0.05$) in 2019/ 20, whereas the 2020/ 21 year produced significant differences ($p < 0.05$). Tree transpiration is known to be influenced by climatic variables; therefore, a Random Forest regression model was used to test the level of influence between tree transpiration and climatic parameters. The soil water content, solar radiation and vapour pressure deficit were found to highly influence transpiration, suggesting these variables can be used in future water-use modelling studies. The profile water content recharge was influenced by rainfall events. After rainfall and soil profile water recharge, there was a rapid depletion of soil water by the *GN* trees, while the soil profile was depleted more gradually at the *P. elliottii* site. As a result, trees at the *GN* site appeared to be water stressed (reduced stem diameters and transpiration), suggesting that there was limited access to alternative water source (such as groundwater). The study concluded that previous long-term paired catchment studies indicate that eucalypts use more water than pine, however, periods of soil water stress and reduced

transpiration observed in this study must be accommodated in hydrological models. Long-term total soil water balance studies

are recommended in the same region to understand the long-term impact of commercial plantations on water resources.

**1 Introduction**

The expansion of new areas of commercial afforestation in South Africa have generally slowed in recent years in favour of the composition of existing plantations changing. This decrease has been attributed to political, environmental and climate change influences (Nambiar, 2019). Pine plantations are still the dominant species in South Africa occupying approximately 49.6%

of total commercial forest plantation areas (Forestry South Africa, 2020). These plantations are mainly grown for sawlog (74.7%) and coarse-fibre pulpwood (24.9%). Over the years, there has been an improvement in the bioeconomy market for dissolving wood pulp products such as short fine-fibre pulp. Fast-growing *Eucalyptus* species are now being considered an alternative to pine due to their superior fibre and pulping properties (Dougherty and Wright, 2012), short rotation (8-12 years) and high productivity rates (Albaugh et al., 2013). *Eucalyptus* plantation productivity can be as high as 35 $m^3$ $ha^{-1}$ $year^{-1}$ on

highly productive sites compared to 25-27 $m^3$ $ha^{-1}$ $year^{-1}$ of pine (Fox et al., 2007). As a result, over the past 10 years, the areas planted to pine in South Africa have decreased by 2% while *Eucalyptus* increased by 10% (Forestry South Africa, 2020). There are now plans to replace as many as 300 000 ha of pine with *Eucalyptus* over the next 20 years (Forestry South Africa, 2020). The potential for an increase in planting *Eucalyptus* species in South Africa may present several environmental considerations including a potential impact to biodiversity (Callaham et al., 2013) and high rates of transpiration and total evaporation

(Stanturf et al., 2013). There is a wide body of knowledge indicating that *Eucalyptus* species transpiration is greater than pine (Scott and Lesch, 1997; Albaugh et al., 2013) and can reduce off-site water yield (Calder, 2002). Given the imminent increase in *Eucalyptus* plantations in the near future, it is vital to understand water-use by pine and *Eucalyptus*. A *Eucalyptus grandis* versus *Pinus patula* comparison by Scott and Lesch (1997) on very deep soils, found that *E. grandis* used up to 100 mm more water per year than *P. patula* using streamflow measurements. In contrast, White et al. (2021), reported no annual differences

between water use of *E. globulus* and *P. radiata* in central Chile.
*Pinus elliottii* and *E. grandis* x *Eucalyptus nitens* clonal hybrid (*GN*) are the second and fourth most planted species in South Africa, respectively. There is no existing literature that quantifies transpiration by these two species in South Africa and there are mixed reports in international literature. The objective of this study was therefore to measure transpiration (as an indicator of tree water-use) of *GN* and *P. elliottii* plantations and the impact posed by each species on plantation water yield. In South

Africa, forest companies harvest trees at an age aimed at maximising profit, with *Eucalyptus* species (grown for pulp) rotation generally ranging from 7 to 10 years, whereas pine (grown for sawlog) is usually 18 to 24 years (Forestry South Africa, 2020). Using an average rotation of 8.5 years for *Eucalyptus* and 21 years for pine, transpiration measurements in our study were conducted on a 8-year-old *GN* (approximately 94% into the rotation) and 20-year-old *P. elliottii* (approximately 95% into the rotation) and both species were therefore presumed to be in the same stage of development (same rotational age).

## 2 Methods

### 2.1 Description of study area

The study area was located on the Mistley Canema estate (29°12'19.78"S, 30°39'3.78"E) in the KwaZulu-Natal midlands of South Africa, which is about 70 km north-east of Pietermaritzburg (Fig. 1). The area is generally hilly with rolling landscapes and a high percentage of arable land (Everson et al., 2014). It is dominated by forb-rich, tall, sour *Themeda triandra* grasslands of which only a few patches remain due to invasion of native *Aristida junciformis*. Soils in this area are highly leached with apedal and plinthic soil forms, mostly derived from shales (Ecca group). The area experiences mist which could significantly contribute to overall precipitation (Mucina and Rutherford, 2006). The study site weather classification according to Koppen-Geiger climate classification (Peel et al., 2007) falls within the Cwa bioclimatic zone characterised by dry, cold winters and warm and wet summer months.

### 2.2 Site description

The study sites were located adjacent to the Two Stream Research Catchment used in previous (Clulow et al., 2011; Everson et al., 2014) forestry research (Fig. 1). Study site 1 was situated on the north-western side of the catchment (1.6 km away) and planted to *GN* in August of 2013. Study site 2 (3.5 km away from the catchment) was established in August 2001 and planted to *Pinus elliottii*. Soil characteristics from the Two Streams research catchment, which is adjacent to our study sites, is presented in Table 2 to show a general soil characteristics picture. Research by Clulow et al. (2011) and Everson et al. (2014) at Two Streams Research Catchment classified the soil profile to be as deep as 13 m (Table 2) and below that consists of a weathered bedrock (saprolite) and fractured basement rock. The soil form was classified as Hutton (Soil Classification Working Group, 1991). Study sites were 4 km away from each other with the automatic weather station located approximately equidistant between the two sites. Both *GN* and *P. elliottii* were planted at a spacing of 2 x 3 m (1667 trees ha$^{-1}$). The *GN* trees were established using cuttings, while for *P. elliottii*, seedlings were used. Both study sites were subjected to standard afforestation practices such as pruning and thinning, weeding prior to canopy closure and slash removal every 5[th] row to minimise fire risk.

### 2.3 Environmental monitoring

An automatic weather station (AWS) was installed on a flat uniform grassland area in the middle of the two study sites to provide supporting meteorological measurements. Measurements of air temperature ($T_{air}$, °C) (HMP 60, Vaisala Inc., Helsinki, Finland), the relative humidity (*RH*, %) (HMP60, Vaisala Inc., Helsinki, Finland), the wind speed (m s$^{-1}$) and direction (degrees) (Model 03003, R.M. Young, Traverse City, Michigan, USA), the solar radiation ($I_s$, MJ m$^{-2}$ day$^{-1}$) (Kipp and Zonen CMP3) and rainfall (mm day$^{-1}$) (TE525, Texas Electronics Inc., Dallas, Tx, USA) were conducted every 10 s and output hourly. The sensors were installed according to recommendations of the World Meteorological Organisation (WMO, 2010) with the rain gauge orifice at 1.2 m and the remaining sensors at 2 m above the ground surface. The CR1000 datalogger (Campbell

Scientific Inc., Logan, Utah, USA) recorded 5-min outputs and was programmed to calculate the Vapour Pressure Deficit (VPD, kPa) using $T_{air}$ and $RH$ measurements according to Savage et al. (1997).

## 2.4 Transpiration flux measurements

Four representative trees were selected within each study site based on diameter stratification. This was achieved by measuring 48 tree diameters at breast height (DBH, 1.3 m) using a diameter tape and stratifying the measured trees according to four size classes; small, medium, medium large and large.

The heat ratio method of a heat pulse velocity system (HPV) (Burgess et al., 2001) was used to estimate sap-flow at various depths across the sap-wood of each selected tree for the 2019/ 20 (October 2019 to October 2020) and 2020/ 21 hydrological years (October 2020 to October 2021). The HPV system consisted of a line heater probe (4 cm long and of 0.18 cm outside diameter brass tubing) with enclosed constantan filament that provides a heat source for 0.5 s when powered and a pair of type T copper-constantan thermocouples to measure the heat ratio. For *Pinus elliottii* trees, slightly longer heater probes (6 cm) were used due to the xylem being situated deeper. Prior to probe installation, thickness of the bark was measured, and suitable sensor insertion depth was identified using an increment borer and Methyl Orange staining. The thermocouples and heater probes were inserted in holes, which were made using a drill and a drill guide to ensure that holes were drilled with the correct spacing and parallel alignment. A heater probe was installed in the central hole and thermocouples installed in each of the holes up (upstream) and down (downstream) from the heater probe relative to the sap-flow direction. Probes were installed at various depths (Table 3) within a tree. Hourly measurements were executed and recorded using a CR1000 datalogger (Campbell Scientific Inc.) powered by a single 55-amp hour lead acid deep cycle battery. Thermocouples were connected to an AM 16/32B multiplexer (Campbell Scientific Inc.), which was in turn connected to a datalogger (CR1000, Campbell Scientific Inc.) to allow for 32 measurements at various sap-wood depths across the four instrumented trees. Data were remotely downloaded using a GSM modem (Maestro Wireless Solutions Ltd. Hong Kong, China)

Hourly measurements started by measuring each thermocouple ten times for accurate initial temperatures. Following a heat pulse, the downstream and upstream temperatures were measured 40 times between 60 and 100s. Thereafter, heat pulse velocity ($V_h$, cm hr$^{-1}$) was calculated using (Burgess et al., 2001),

$$V_h = \frac{k}{x} \ln \left( \frac{V_1}{V_2} \right) 3600 \tag{1}$$

Where $k$ is a thermal diffusivity of fresh wood (a nominal value of 2.5 x 10$^{-3}$ cm$^2$ s$^1$), $x$ is the distance of each temperature probe from heater probe (cm), and $V_1$ and $V_2$ are temperature increases in upstream and downstream probe ($^o$C) at equidistant points.

A slight probe misalignment may occur during the drilling process even when a drill guide is used. This was assessed by checking for inconsistencies in the zero flux values in periods where sap-flow was expected to be zero, such as over pre-dawn, during rainfall events, or in high $RH$ and low soil water content ($SWC$) conditions. The sap-flow values during these times

were adjusted to zero and an offset calculated from an average of these values and applied to the whole dataset. For probes used in this study, the offset was < 5% of the midday sap-flow rates.

Wounding or non-sap conducting area around the thermocouples was accounted for using wound correction coefficients described by Burgess et al. (2001). Thereafter, sap velocities were calculated accounting for moisture fraction and wood density as described by Burgess et al. (2001). Finally, sap velocities were converted to transpiration rates (mm day$^{-1}$) by summing products of sap velocity and cross-sectional sapwood area for individual stems. The transpiration rates were then weighted as per individual tree contribution to provide a measure of whole stand transpiration.

**2.5 Soil water content measurements**

At both sites hourly *SWC* was measured in the upper 0.60 m of the soil profile (0.20 m, 0.40 m and 0.60 m depth) using CS616 soil water measuring sensors (Campbell Scientific Inc.). The CS616 *SWC* sensor consists of two 30 cm long stainless-steel rods that uses the time domain reflectometer method to measure the *SWC*. The sensor circuitry generates an electromagnetic pulse, of which an elapsed pulse travel time and reflection are measured and then used to calculate the *SWC*. Research

conducted at Two Streams Catchment (Clulow et al., 2011; Everson et al., 2014) reported that the majority of large and fine roots were located in the top 0.06 m to 0.4 m of the soil profile, hence *SWC* measurements were conducted in the top 0.6 m of the soil profile in our study. The *SWC* measurements ran concurrently with the sap-flow measurements and were recorded on a datalogger (CR1000, Campbell Scientific Inc). The profile water content at 0.6 m soil depth ($PWC_{0.6}$, mm of water per 0.6 soil depth) was estimated from the *SWC* measurements using:

$$PWC_{0.6} = \left(\frac{(SWC_{0.2} \times 0.2) + (SWC_{0.4} \times 0.2) + (SWC_{0.6} \times 0.2)}{1000}\right) \times 100 \qquad (2)$$

where, $SWC_{0.2}$, $SWC_{0.4}$, $SWC_{0.6}$ was the soil water content measured at 0.2 m, 0.4 m and 0.6 m, respectively.

**2.6 Heat ratio technique calibration**

The HPV method is an internationally recognised and reliable technique for measuring individual tree transpiration in uniform stands (Hatton and Wu, 1995; Meiresonne et al., 1995; Crosbie et al., 2007). There are however some uncertainties to the accuracy of the absolute sap-flow results, such as the anisotropic sap-wood properties (Vandegehuchte et al., 2012), radial patterns of the sap-flow (Cermák and Nadezhdina, 1998), tree symmetry (Vertessy et al., 1997) and changes in spatial patterns of transpiration (Traver et al., 2010). Some studies have indicated that the technique underestimates sap-flow in *Eucalyptus* by

as much as 45% (Maier et al., 2017; Fuchs et al., 2017), whereas pine may be overestimated by as much as 49% (Dye et al., 1996b). This necessitated a calibration experiment to validate the field measurements.

The calibration experiment was conducted in an open area at the Institute for Commercial Forestry Research nursery, located at the University of KwaZulu-Natal, Pietermaritzburg for a period of 30 days as illustrated in Figure 2. Two-year-old *GN* and four-year-old *Pinus elliottii* trees grown in 25-L plastic containers (diameter=36 cm, height = 42 cm) filled with vermiculite were sourced from Mondi Mountain Home Estate nursery (Hilton, South Africa). The containers had holes at the base (to allow for drainage) and were placed on a rubber mat with slots to prevent root contact with soil and to allow water to drain away from the container. Twenty-four hours before starting the experiment, both trees were well watered, and each container was insulated using plastic at the tree base to prevent soil evaporation and induce water loss solely through transpiration. Tree diameters at the start of an experiment were 0.044 and 0.036 m for *GN* and *P. elliottii*, respectively. Each tree was instrumented with HPV sensors to measure hourly sap-flow (as discussed in section 2.4) and summed from sunrise to sunset to make up daily tree transpiration (L day$^{-1}$). Concurrently, each soil container was weighed in the morning and afternoon, using a lysimeter (resolution=0.001g, placed on a flat concrete surface to ensure it remains level during the experiment) to determine daily changes in container weight (kg, where 1 kg was assumed to be equivalent to 1 L) as a measure of transpiration. This process was repeated for five days to get a calibration over a range of plant available water values, whereafter trees were again well-watered (achieved by removing insulation plastic) and allowed to drain completely before restarting measurements. Sapwood area and wounding was accounted for according to Burgess et al. (2001) to derive daily transpiration. A simple regression was conducted between daily transpiration and daily change in tree mass.

## 2.7 Growth measurements

Measurements of DBH were conducted monthly using manual dendrometer bands (D1, UMS, Muchin, Germany) permanently attached to a tree, with an accuracy of 0.1 mm. Dendrometer bands were installed at beginning of October 2019 on 48 trees including the four HPV instrumented trees and data were manually collected for 21 months. The quadratic mean diameter ($D_q$) was calculated for 48 trees using (Curtis and Marshall, 2000):

$$D_q = \sqrt{\frac{\sum(d^2)}{n}} \tag{3}$$

where; $d$ is the DBH (m) of an individual tree and n is the total number of trees. Tree heights for the 48 trees were measured simultaneously using a hypsometer (Vertex Laser VL402, Haglof, Sweden). Monthly measurements of leaf area index were conducted using a LAI-2200 Plant Canopy Analyzer (Licor Inc., Lincoln, New York, USA) on a 0.6 ha transect that was identified through the middle of each study site from October 2019 to October 2021.

## 2.8 Statistical analysis

Analysis of variance (ANOVA) was used to analyse species differences in stand characteristics (transpiration, $D_q$, tree heights and leaf area index) using the R version 3.6.1 statistical package. Variables were transformed as appropriate to meet the assumptions of normality. Where the overall F-statistic was significant ($p < 0.05$), treatment means were compared using Fischer's Least Significant Difference at the 5% level of significance (LSD$_{5\%}$). Statistical parameters that were used included

the regression co-efficient ($R^2$), root mean square error (RMSE), standard error of a regression slope (SE slope), standard error of the intercept (SE intercept) and a ratio of variance of y-intercept to x-intercept (F). In addition, Random Forests (RF) regression algorithm (Breiman, 2001) in R statistical computing software (R Development Core team, 2008) was used to rank climatic variables that influence transpiration, where transpiration was made a response variable and meteorological data ($I_s$, VPD, *SWC*, $T_{air}$, rainfall, wind speed and RH) as predictor variables. This machine learning approach does not make the assumptions of linear regression and performs well when the relationship among the response variable and independent variables are complex and non- linear. The RF regression model was optimised in terms of the parameters *ntree* (number of trees built by the model) and *mtry* (number of variable predictors used at each node split using the Caret package) (Kuhn, 2008). The RF regression was evaluated using the $R^2$ metric and the contribution of each variable to the model accuracy was determined by developing a variable importance plot. The variable importance was calculated from the out-of-bag (OOB) samples. Using a bootstrap sample with replacement, two thirds of the original dataset are used to train individual trees in the ensemble, whereas the remaining one third of a sample is used for determining ranked variable importance, providing a measure of accuracy (Breiman, 2001). In this study, the two thirds of the dataset for each measurement period were used for calibrating and validating the model, while the one third was used for testing the model. The variable importance plot was assessed using the mean decrease accuracy (MDA) coefficient measures (Breiman et al., 1984). The MDA is calculated during the OOB sample computation phase. The values of a particular variable are randomly permuted on the OOB sample, enabling the new classification to be determined from the modified sample. For more details on how MDA is quantified refer to Cutler et al (2007) and Aria et al (2021). The difference between the rate of misclassification for the modified sample and the original sample is used as a measure of the variable importance. Each predictor variable was scored based on the MDA for *GN* and *P. elliottii* measurement period.

## 3 Results

### 3.1 Automatic weather station

The minimum and maximum daily $T_{air}$ were typical of the 30-year average of Mistley Canema. Maximum recorded $T_{air}$ was 36.5 and 37.5°C for 2019/ 20 and 2020/ 21 hydrological years, respectively. There were several days where $T_{air}$ were below freezing between May and July for both measurement years (Fig. 3a). Rainfall between 01 October 2019 and 30 September 2020 amounted to 857 mm and 825 mm for 01 October 2020 to September 2021. Majority of this rainfall (70%) fell during summer months (November to March) for both years (Fig. 3d). By comparison, potential evaporation totals calculated using hourly AWS data and the FAO56 method (Allen et al., 1998) amounted to 1100 mm and 1056 mm for 2019/ 20 and 2020/ 21 hydrological years, respectively. Daily maximum VPD was 3.08 kPa for 2019/ 20 increasing to 3.53 kPa for 2020/ 21 year during hot summer months (Fig. 3c). Monthly average wind speed ranged from 2.2 to 7.7 m s$^{-1}$ over the two hydrological years with maximum wind speeds up to 37 m s$^{-1}$ in August/ September. The *RH* reached 100% during the night, decreasing to as

low as 20% during the day on hot summer months. Average $I_s$ for 2019/ 20 and 2020/ 21 hydrological years was 15.5 and 16

MJ m$^{-2}$ day$^{-1}$, respectively, while both years experienced a maximum $I_s$ of 30 MJ m$^{-2}$ day$^{-1}$ in summer (Fig. 3b).

### 3.2 Soil profile water content

The $PWC_{0.6}$ was very responsive to rainfall events (Fig. 4 and Fig. 7a) on both study sites. The peak $PWC_{0.6}$ for the $GN$ site during the wet season was 227 mm and 198 mm day$^{-1}$ in 2019/ 20 and 2020/ 21 hydrological years, respectively. By

comparison, the maximum measured $PWC_{0.6}$ in the *P. elliottii* site was 128 mm day$^{-1}$ in 2019/ 20 and 125 mm day$^{-1}$ in 2020/ 21. The $PWC_{0.6}$ at both study sites did not significantly respond to rainfall events below 5 mm hr$^{-1}$, except during consecutive rainfall events. After a significant rainfall event, the $PWC_{0.6}$ for the $GN$ site was depleted rapidly, within hours (Fig. 4 and Fig. 7a), which contrasts with the *P. elliottii* site, where $PWC_{0.6}$ was depleted more gradually, lasting for a few days after the rainfall event. The swift depletion of plant available water at the $GN$ site, resulted in the site experiencing extended periods of low-

profile water content. During the dry season, the $PWC_{0.6}$ was maintained at approximately 50 and 60 mm day$^{-1}$ for *P. elliottii* and $GN$ (Fig. 7b), respectively, except when significant rainfall events occurred.

Commercial forest plantations are known to have a very deep rooting system and are able to access soil water in deeper soil water reserves (Christina et al. 2016). A study adjacent to our study site (Everson et al. 2014) reported that *Acacia mearnsii*

tree roots were as deep as 8 m into the soil profile. Similar results were reported by Dye (1996) in the Mpumalanga province of South Africa, where *Eucalyptus grandis* trees abstracted water down to 8 m below the soil surface. The deep soil profile with the presence of weathered bedrock (saprolite) in our study site suggests that trees were capable of rooting as deep as 20 m into the soil profile and were probably restricted by the bedrock (grey fine-grained shale). However, Hasenmueller et al. (2017) indicated that shale may consist of fractures where tree roots may grow through. There is, therefore, a high possibility

that tree roots in this study accessed soil water stored deep in the soil profile from previous wet years.

### 3.3 Heat ratio calibration

The HPV system slightly overestimated transpiration (in the case of the $GN$) and underestimated transpiration (in the case of the *P. elliotii*) when compared to a lysimeter system. A simple regression between the two systems produced a good linear relationship (*GN*: R$^2$=0.73, *P. elliottii*: R$^2$=0.76) for both tree species (Fig. 5a and 5b), with a RMSE of 0.57 and 0.36 L day$^{-1}$

for the $GN$ and *P. elliottii*, respectively. This relationship was used to correct the transpiration results for both tree species:

$$GN = 1.17x - 0.011 \tag{4}$$
$$P.\ elliottii = 0.81x + 0.11 \tag{5}$$

### 3.4 Transpiration rates

The transpiration followed typical seasonal and diurnal pattern for both sites in both 2019/ 20 and 2020/ 21 hydrological years (Fig. 6). *Pinus elliottii* had significantly ($p < 0.01$) higher mean daily transpiration compared to *GN* (Fig. 6) except for the winter of 2021 (May to August) where *GN* was statistically ($p=0.012$) greater. Mean daily transpiration values in summer of 2019/ 20 for *Pinus elliottii* and *GN* were 2.5 and 1.9 mm day$^{-1}$, respectively. By comparison, summer mean transpiration values of 2020/ 21 were 2.6 mm day$^{-1}$ for *P. elliottii* and 2.1 mm day$^{-1}$ for *GN* ($p < 0.05$). After a significant rainfall event (~5 mm hr$^{-1}$),

transpiration for *GN* momentarily exceeded *P. elliottii* for a few days, thereafter, falling below *P. elliottii* again. The maximum transpiration for *GN* was 5.2 mm day$^{-1}$ and 3.8 mm day$^{-1}$ for 2019/ 20 and 2020/ 21 measurement year, respectively, versus 5.6 mm day$^{-1}$ for *P. elliottii* in both seasons. During 2019/ 20, *GN* reached peak transpiration rates early in summer (late December 2019) compared to *P. elliottii*, where peak transpiration rates were measured in late January to early February of 2020 (Fig. 6). However, maximum transpiration rates were reached mid-January for the 2020/ 21 measurement year by both

crops, which coincided with high $I_s$, $T_{air}$ and VPD. During winter months (June to July) of both the 2019/ 20 and 2020/ 21 hydrological years no transpiration could be detected by probes on *GN* trees on several days, despite clear weather conditions. This corresponded with low $PWC_{0.6}$ (approximately 60 mm per 0.6 m soil depth). By comparison, transpiration could be measured in *P. elliottii* trees where the $PWC_{0.6}$ was low (~ 50 mm day$^{-1}$ per 0.6 m soil depth), although at very low transpiration rates (~0.33 mm day$^{-1}$). Following rainfall, the *P. elliottii* response to *PAW* lagged behind the *GN* trees. While *GN* transpiration

increased almost immediately, *P. elliottii* transpiration only responded a few days later (Fig. 7a and Fig. 8).

The differences in seasonal patterns of transpiration are illustrated using daily accumulated transpiration (Fig. 9). Over the 2019/ 20 measurement year, the total accumulated daily transpiration for *P. elliottii* was 30% greater than *GN*. The total accumulated transpiration rate of *P.elliottii* was also higher in 2020/ 21 but statistically similar *(p > 0.05)*. Total annual

transpiration rates for *GN* were slightly higher in 2020/ 21 than 2019/ 20 measurement years (6%), while *P. elliottii* transpiration rates reduced by 19% over the same period (Fig. 9). The accumulated rainfall was 18 and 20% greater than transpiration for *P. elliottii* and *GN*, respectively, while the accumulated potential evaporation exceeded rainfall by 22% in both seasons.

### 3.5 Influence of climatic variables on tree transpiration

The RF model performed well in ranking climatic variables influencing transpiration for both species, producing an overall coefficient of determination ($R^2$) of 0.91 and 0.85 for *GN* and *P. elliottii*, respectively. The overall root mean square error was 0.58 mm for *GN* and 0.9 mm for *P. elliottii*, indicating a very good predictive power. The RF predictive model rated *SWC* measured at 40 cm depth and *SWC* measured at 60 cm depth as the most important influencers of transpiration for *GN* and *P. elliottii*, respectively (Fig. 10). Climatic variables, $I_s$, VPD and $T_{air}$ in descending order of importance were also scored as

important in *GN*, whereas in *P. elliottii*, soil water content measured at 40 cm, VPD and $I_s$ were found to be important variables

(Fig. 10). Rainfall was found to be the least important variable in a model in *P. elliottii*, while wind speed was not a good influencer of transpiration in *GN*. The model indicated that transpiration is influenced by micrometeorological variables at varying levels of influence.

### 3.6 Tree growth

At the beginning of the study, *P. elliottii* trees were larger in diameter than *GN*. There was a seasonal pattern in $D_q$ increment by both species (Fig. 11), with no significant ($p > 0.05$) differences in 2019/ 20, while 2020/ 21 produced significantly ($p < 0.05$) greater growth increment in *P. elliottii* than *GN*. Interestingly, a negative growth increment was measured during the winter of 2019/ 20 for *GN*, which was probably caused by low $PWC_{0.6}$.

### 3.7 Leaf area index

The mean summer leaf area index for *P. elliottii* was 17% greater than *GN* (*P. elliottii*=2.5 vs *GN*=2.05, $p < 0.05$) in 2019/ 20 increasing to 21% (*P. elliottii*=3.1 vs *GN*=2.4, $p < 0.05$) in 2020/ 21. Winter leaf area index decreased to 1.31 and 1.76 for *P. elliottii* and *GN*, respectively. Total monthly transpiration was linearly related to monthly leaf area index of both *P. elliottii* and *GN* (Fig. 12), with statistical differences in the regression ($p < 0.05$). However, there was a greater RMSE, SE intercept and SE slope in *P. elliotti* than in *GN*, indicating that the regression line in *GN* fits the data better than *P. elliottii*, and *GN* has

more precise prediction of transpiration from leaf area index than *P. elliottii*.

## 4 Discussion

### 4.1 Daily transpiration

    The *P. elliottii* mean daily transpiration exceeded *GN* in 2020/ 21 measurement years, mainly influenced by *SWC*, VPD and $I_s$. Differences in transpiration between *GN* and *P. elliottii* could be attributed to the following reasons: 1) trees at the *GN* site

were water stressed and evidence of water stress was observed through shrinking of tree stem diameters during winter, zero rates of transpiration on some days during winter months and a significant decrease in leaf area index over winter. This suggests that trees were unable to access soil water stored from previous wet years held deep in the soil profile or the *GN* trees had already accessed and depleted the stored soil water before the study period. *Eucalyptus* transpiration has been shown to increase sharply in the early stages of growth, reaching a peak in the middle of the rotation, thereafter, declining as the stand matures

(Delzon and Loustau 2005) and, 2) sap-wood for *P. elliottii* was nearly twice the sap-wood area of *GN* due to the different tree structures. The *GN* mean transpiration range of 0.9–5.2 mm day$^{-1}$ and 0.5–3.8 mm day$^{-1}$ for 2019/ 20 and 2020/ 21, respectively, measured in this study agreed with *Eucalyptus* studies in relatively low rainfall areas with trees of the same age. For example, a study by Forrester et al. (2010) on seven-year-old *E. globulus* in Australia measured a transpiration range of 0.4–1.9 mm day$^{-1}$ (MAP=708 mm). David et al. (1997) measured daily transpiration of 0.5–3.64 mm day$^{-1}$ at a *E. globulus* site in Portugal

with a MAP of 600 mm. A South African study by Dye et al. (1996a) on nine-year-old *E. grandis* in Mpumalanga, South

Africa measured transpiration of 2.0–7.5 mm day$^{-1}$ with the potential to exceed 8.0 mm day$^{-1}$ under high VPD (Dye et al., 2013), however, this study was conducted in a high rainfall area (MAP=1459), with almost double the MAP of the current study. For *P. elliottii*, peak transpiration of 5.6 mm day$^{-1}$ in this study agreed with other studies, such as Hatton and Vertessy (1990) who measured a maximum transpiration of 5 mm day$^{-1}$ in *P. radiata* in new South Wales, Australia.

During summer, *GN* transpiration peaked earlier than the *P. elliottii* (more distinct in 2019/ 20) and then decreased swiftly, so that it was less than the *P. elliottii* transpiration in the late summer to early autumn. In addition, the *GN* transpiration increased sharply after the rainfall events and thereafter decreased as $PWC_{0.6}$ was rapidly depleted, while *P. elliottii* responded more gradually. This suggests that *GN* trees have a different growth and water-use strategy to *P. elliottii,* that involves using available water rapidly. A similar observation was reported by White et al. (2021) from *E. globulus* in central Chile. This implies that

*GN* trees compete for water and use it more rapidly when it becomes available, and this strategy can expose them to extreme water stress if soil water deficit conditions persist as reported by Mitchell et al. (2013). *P. elliottii* had a greater transpiration and stem sizes similar to the *GN* (ie. $D_q$ for smallest *P. elliottii* tree versus $D_q$ for largest *GN* tree). This may be attributed to a markedly smaller heartwood in *P. elliottii* than *GN*. However, it should be noted that pine trees consist of several latewood rings in which no sap movement occurs (Dye et al., 2001). Diurnal changes in transpiration typically lagged behind VPD,

creating a pattern of hysteresis, where at similar VPD, transpiration was greater in the morning than in the afternoon for *GN*. Studies by O'Grady et al. (1999) and Maier et al. (2017) attributed this to low soil hydraulic conductivity or the use of stored stem water for transpiration in the first portion of the day. Further analysis in our study indicated that *GN* transpiration was significantly influenced by VPD only in summer, suggesting that soil water deficit may have affected soil water uptake to a greater extent in the dry season (winter).

**4.2 Annual T**

  On an annual basis, *P. elliottii* trees transpired 28% more water (836 mm) than *GN* (599 mm year$^{-1}$) in 2019/ 20, while the 2020/ 21 saw no significant differences between the two species (*P. elliottii*=678 mm year$^{-1}$ vs. *GN*=639 mm year$^{-1}$). The low rates of transpiration in winter months (May to August) of 2021 on the *P. elliotti* site (Fig. 6), were caused by low *SWC*, which resulted in similar annual transpiration rates in 2020/ 21 by both species. Other studies of pines (Moran et al., 2017; Samuelson

et al., 2019) reported that the first reaction by pine species to a decrease in *SWC* is a significant reduction in stomatal aperture, causing a decrease or cessation of transpiration. By comparison, *GN* indicated a different response, where transpiration continued (even when *SWC* was marginally limiting) to a point where it was below detection by our HPV system and this trait makes eucalypts vulnerable during extended or severe drought periods. South African studies (Dye et al. 1996 and Eksteen et al. 2013) using *E. grandis* crop showed that trees utilised *SWC* till permanent wilting point, which seems to be well beyond

the -1500 kPa typically cited in literature for many plants (Santra et al., 2018). An international study by White et al., (1999) found similar results in *E. nitens* in western Australia.

  There are contrasting results in some annual comparative studies of transpiration between *Eucalyptus* and *Pinus* species. In an eight-year-old *E. benthamii* vs *P. taeda* comparative water-use study in the United States (Maier et al., 2017), annual

transpiration of 1077 and 733 mm year$^{-1}$ for *E. benthamii* and *P. taeda*, respectively, were measured. In a South African study

(tree water-use estimated using water balance), *Eucalyptus grandis* used 100 mm more water per year than *Pinus patula* (Scott and Lesch, 1997). Notwithstanding these findings, another study in southeastern Australia, Benyon and Doody (2015) found no significant differences between annual water-use (only transpiration was measured) between *E. globulus* and *P. radiata*, with or without access to groundwater. A most recent study by White et al. (2022) using meta-analysis of published evapotranspiration estimates found no significant differences in water-use between *Eucalyptus* and *Pinus* genera. Like other

studies (Whitehead and Beadle, 2004; Samuelson et al., 2008), a strong correlation between transpiration and tree leaf area was observed in this study, with *P. elliottii* having a greater transpiration rate than *GN* at a similar leaf area index. This good correlation may present a modelling opportunity to estimate transpiration using site measurements of leaf area index or remote sensing estimates of leaf area index.

**4.3 Response of tree transpiration to climatic variables**

Conventional micrometeorological techniques of estimating transpiration (such as heat pulse velocity) in commercial forest plantations are not easy to conduct due to the remote and inaccessible nature of forest plantations, vulnerability of instrumentation to damage by animals and a threat posed by extreme weather events. In addition, this instrumentation is very expensive, requires technical skills to use and provides measurements on few trees within a stand. An improved technique for estimating transpiration from easy to measure variables would be an advantage. There are many external regulators that have

been described to have a strong relationship with transpiration, which includes readily available soil water in the rooting area (Oren and Pataki, 2001), the atmospheric micrometeorological conditions (Lundblad and Lindroth, 2002) and aerodynamic resistance (Hall, 2002). However, these relationships are complex, because exotic trees can have several internal mechanisms, which can vary between species, tree age and tree physiology (Zweifel et al., 2005). Nevertheless, in most actively growing tree species, there is a consensus that certain meteorological variables can influence transpiration (Albaugh et al., 2013).

Results from regression analysis using RF showed that *SWC*, VPD and $I_s$ have a significant influence on transpiration, in a decreasing order of importance for both species. These results were expected as other studies have shown that tree transpiration is influenced by *SWC* (Dye, 1996; Maier et al., 2017), VPD (Dye and Olbrich, 1993, Scott and Lesch, 1997; Campion et al., 2004, Albaugh et al., 2013) and $I_s$ (Zeppel et al., 2004; Albaugh et al., 2013). For example, in South African studies by Dye (1996a) and Dye et al (2004), *E. grandis* daily water-use exceeded 90 litres day$^{-1}$ (equivalent to 7 mm day$^{-1}$) on hot dry days

in the middle of summer (when *SWC* was not limiting), and then declined during the dry season as temperatures decrease and the photoperiod shortened. Rainfall was found to be a weak influencer of tree transpiration, which could be linked to high rate of rainfall interception reported in commercial forests of 14.9% for *Eucalyptus* and 21.4% for pine of gross precipitation (Bulcock and Jewitt, 2012). These results suggest that certain climatic variables can be used as an input in commercial forest plantation models to improve estimation of tree water-use.

## 4.4 Implications of tree water-use on water yield

Forest plantation water-use studies and their potential impact on water resources are complex and require comprehensive long-term measurements of total water balance parameters to be conclusive. Our results indicated that *P. elliottii* water-use was significantly greater than *GN* in 2019/ 20, while 2020/ 21 water-use patterns were statistically similar. These year-on-year differences have been attributed to low *SWC* and plant stress reducing the accumulated *GN* transpiration. However, this result is in contrast with most of the long-term studies which indicate *GN* to use more water than *P. elliottii*. It is clear that the implications over a full rotation or over several rotations cannot be quantified from such short-term studies. In particular, lags that occur in hydrology between rainfall, changes in groundwater and streamflow have been reported to exceed the time period of this study and will be poorly captured. There are several long-term paired catchment studies conducted in South Africa (van Lill et al., 1980; Smith and Scott, 1992; Scott and Lesch, 1997; Scott and Smith, 1997; Scott et al., 2000) that compared water-use between pine and *Eucalyptus* and quantified the impact of these species on water resources, particularly the streamflow. A study by Scott and Lesch (1997) on the streamflow response to afforestation with *E. grandis* and *P. patula* in Mokobulaan experimental catchment in South Africa indicated that eucalypts cause a faster reduction in streamflow (90–100%) compared to afforestation with pines (40–60%). These results were verified in a study conducted by Scott et al (2000) where peak reductions in streamflow were reported between 5 and 10 years after establishing eucalypts, and between 10 and 20 years after planting pines, with the size of the reduction driven by soil water availability. Another South African study by Smith and Scott (1992), investigated the impact of pine and eucalypts on low flows in various paired catchments located in various regions of South Africa (Westfalia Estate, Cathedral Peak, Jonkershoek, Mokobulaan). Results from this study showed that afforestation have a significant effect on low flow on all paired catchments (low flows reduced by up to 100% in certain cases), with eucalypts having a severe impact compared to pine. Results from our *GN* study site indicated that the site was likely water stressed, suggesting that trees were probably unable to access soil water stored from previous wet years held deep in the soil profile. Afforestation with commercial forest plantations over successive rotations have been shown to deplete soil water reserves within the profile, leading to increased soil water deficit (Dye et al., 1997) and ultimately reduction in streamflow. Evidence of this was shown in a paired catchment experiment by Scott and Lesch (1997) where eucalypts were clear-felled at 16 years of age, but full perennial streamflow returned five years later. The delay in streamflow recovery was attributed to eucalypts desiccating the deep-water reserves, which had to be restored before the stream could return to a normal flow.

All of the above-mentioned South African long-term paired catchment studies suggest that commercial forest plantations, particularly *Eucalyptus*, pose a severe negative impact on water yield. Results from our study indicated that *P. elliottii* used more water than *GN* over the first year of measurement due to limited soil water availability and a conclusive impact on water reserves cannot be quantified without long-term measurements of at least one crop rotation. This relatively short-term study did however show that commercial forest plantations may deplete soil water stored within the soil profile during dry period, resulting in potential streamflow reduction over a long term. Due to climate variability in plantation forest areas, long-term

studies under non-stressed and stressed conditions are needed in this region to quantify the total water balance (total evaporation, surface runoff, soil water storage and how water partitioning responds to climate change and afforestation over time).

## 5 Conclusions

This paper presents a water-use study by *GN* and *P. elliottii* near the Two Streams Research Catchment in the KwaZulu-Natal midlands of South Africa, quantified using the heat ratio method (HRM). A calibration of the method was conducted and is recommended for this technique to achieve improved accuracy. Annual water-use results indicated that *P. elliottii* used 28% more water than *GN* in the first measurement year (2019/ 20), while there were no significant differences in tree transpiration in the second year of measurement (2020/ 21). These findings contrast with most long-term paired catchment studies in South Africa and internationally, which reported that *Eucalyptus* species are heavy water users compared to pine and both species cause negative impacts on the water yield. This relatively short-term study showed the different responses of the tree species to changes in season and available soil water with *GN* generally responding more rapidly. It also showed that in countries such as South Africa, where streamflow reduction by commercial forestry is modelled for water licensing purposes, soil water stress in the hydrological models must be able to constrain tree water-use. Long-term research is suggested to quantify the total water balance (total evaporation, surface runoff, soil water storage and how water partitioning responds to climate change and afforestation over time), so that the impact of species (such as *GN* and *P. elliottii*) on water yield can be determined. A good relationship between tree transpiration and meteorological variables suggests that "easy to measure" weather variables can be incorporated in future water-use modelling studies to estimate a difficult to derive tree transpiration.

## 6 Data availability

Due to the high-frequency data used for this paper, all data with linked figures and tables have been uploaded to the central database at the Centre for Water Resources Research (CWRR) at the University of KwaZulu-Natal in Pietermaritzburg. The author, Nkosinathi David Kaptein, can be contacted for these data at kapteinnd@gmail.com.

## 7 Author contribution

MLT and ADC were responsible for funding acquisition, resources and project administration. MLT, ADC and CSE conceptualised the study and conceived the methodology. NDK was a student who collected the data, analysed, interpreted, and wrote the original draft of the paper. ADC, MLT, CSE and IG provided student supervision, discussed the results, and contributed to the final version of the paper.

## 8 Competing interests

The contact author has declared that none of the authors has any competing interests.

## 9 Acknowledgements

This research was funded by the Department of Water and Sanitation through Water Research Commission. Mondi Group and Mistley Canema Estate are acknowledged for their support in providing access to research sites. David Borain and Heyns Kotze provided invaluable assistance and support, without which this study would not have been successful. Assistance from Dr Steven Dovey, Mxolisi Gumede, Xolani Colvelle, Jimmy Nhlangulela was much appreciated.

## 10 Financial support

This research has been supported by the South African Department of Water and Sanitation through a Water Research Commission project K5/2791.

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

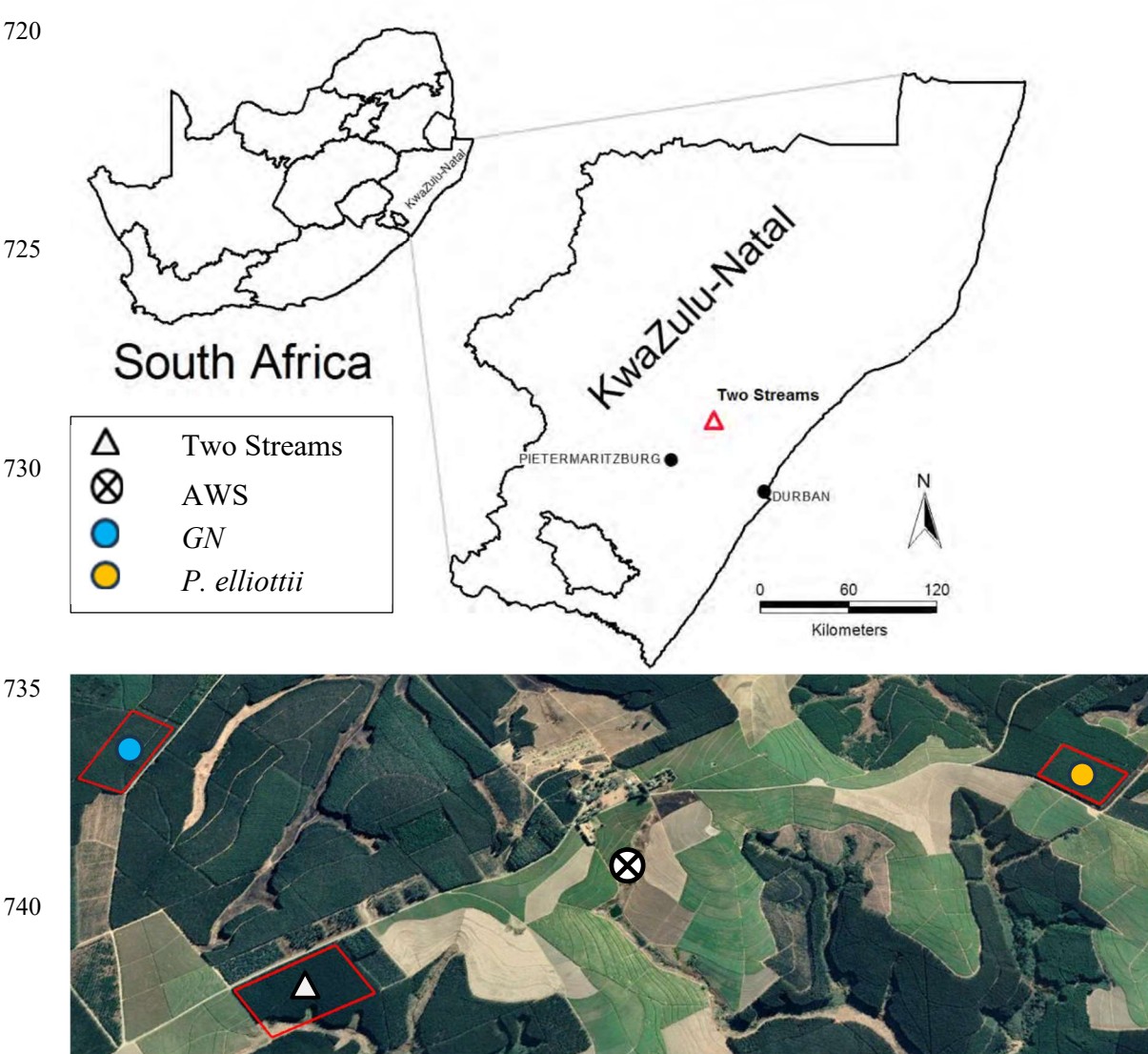



**Figure 1.** Location of the study area next to Two Streams Research Catchment. The Google Earth Pro extract (above) provides location of the two study sites, *E. grandis* x *E. nitens* (*GN*), *Pinus elliottii* and the automatic weather station (© Google Maps 2022). Red lines indicate the boundaries of the compartment for each study site. Dark green vegetation indicates commercial forest plantations and light green vegetation is the sugarcane fields.





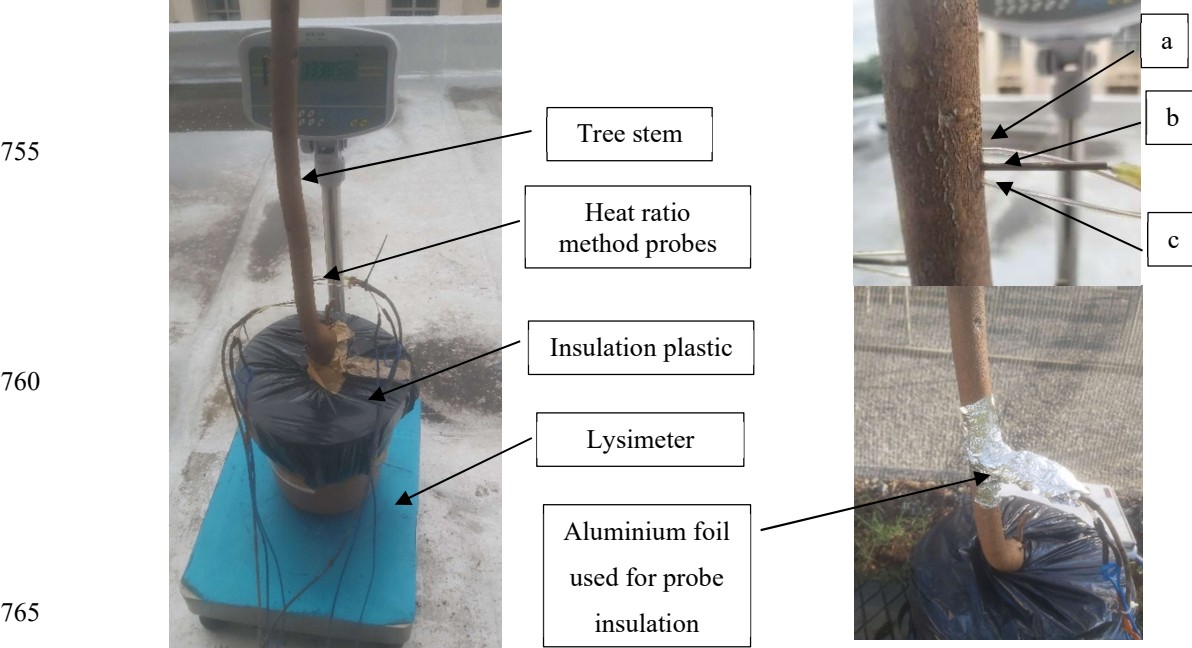

Tree stem

Heat ratio
method probes

Insulation plastic

Lysimeter

Aluminium foil
used for probe
insulation

a

b

c

**Figure 2**. An illustration of a calibration experiment setup showing a tree installed with the heat ratio probes, placed in a lysimeter. Insert: a= downstream probe, b= heater probe, c= upstream probe with aluminium foil used for insulation.











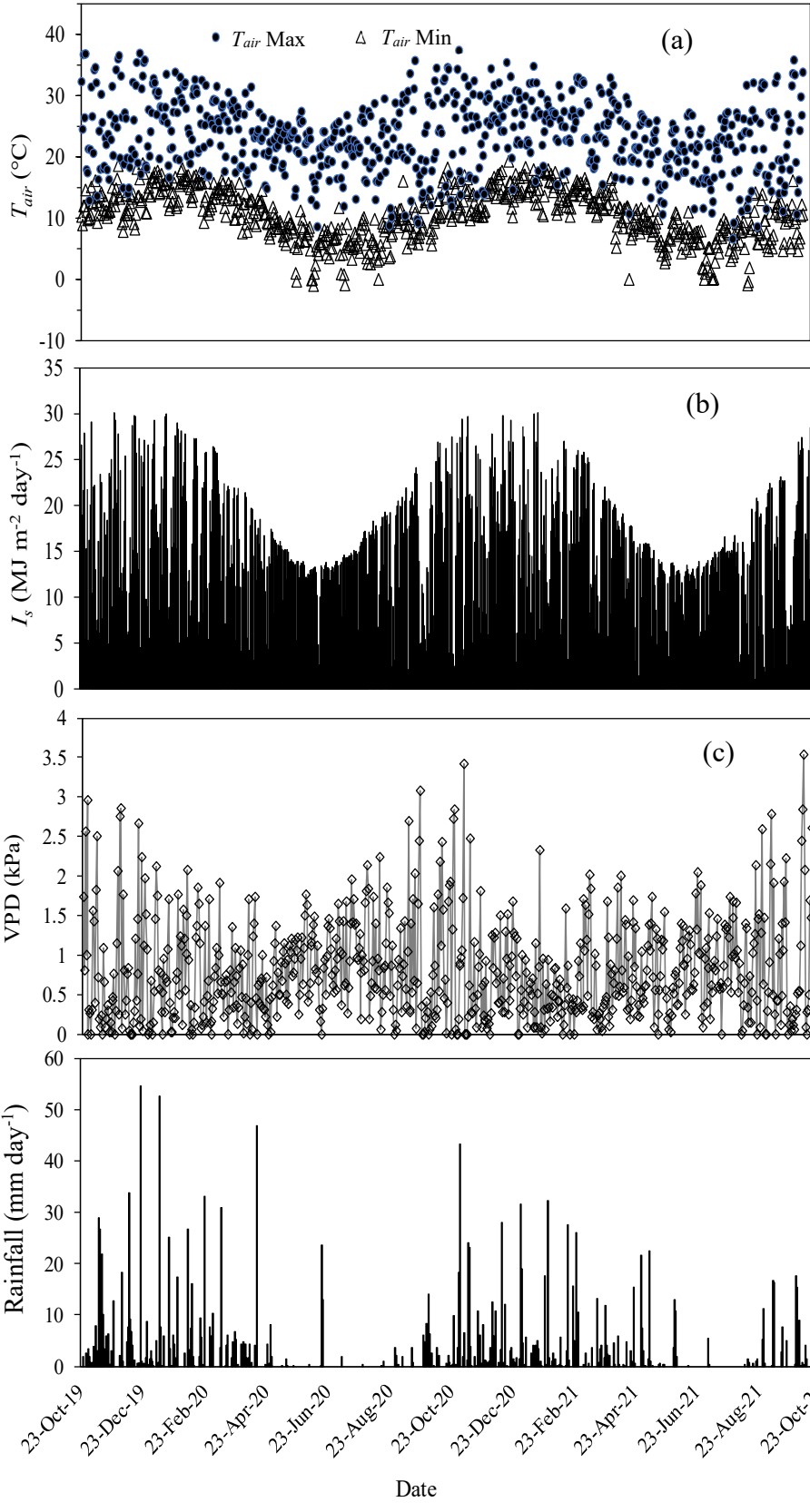

**Figure 3.** (a) the daily minimum ($T_{air}$ Min) and maximum ($T_{air}$ Max) air temperature (°C) (b) daily total solar irradiance ($I_s$, MJ m$^{-2}$ day$^{-1}$) (c) daily mean vapour pressure deficit (VPD, kPa) and (d) total daily rainfall (mm day$^{-1}$) for a duration October 2019 to October 2021.

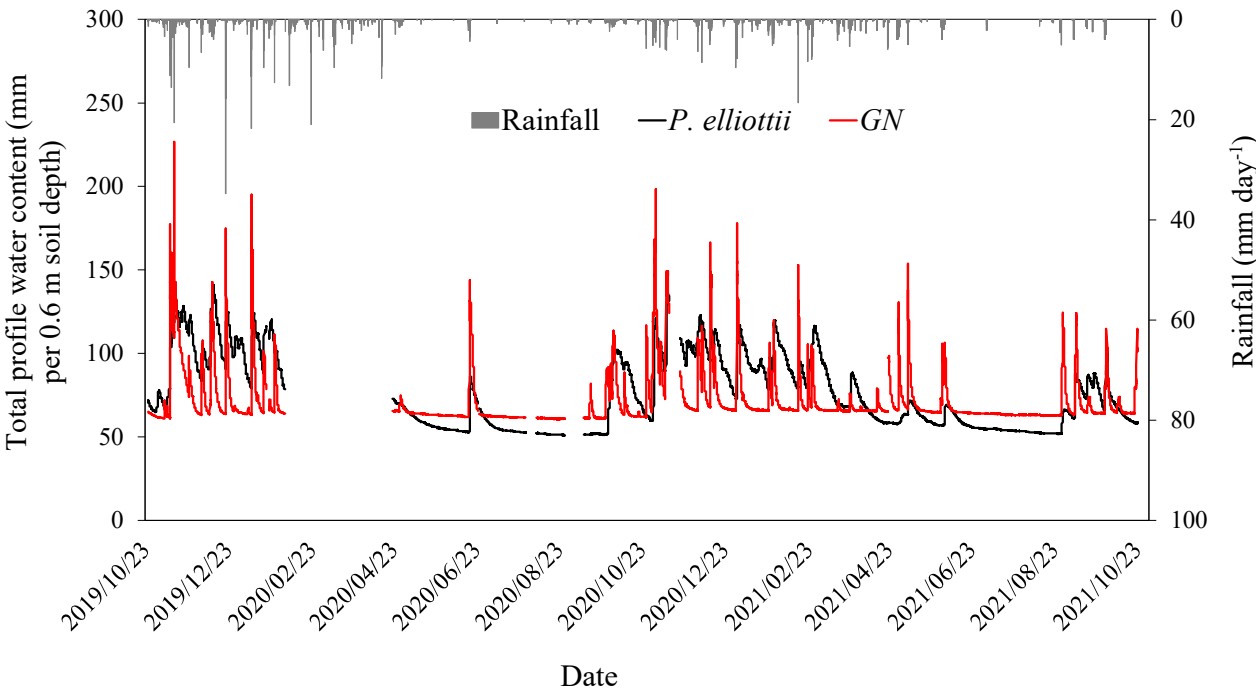

Date

**Figure 4.** The total profile water content (mm day$^{-1}$) measured in the top 0.6 m of the soil profile in the *Pinus elliottii* and *Eucalyptus grandis* x *E. nitens* clonal hybrid sites in response to rainfall events (mm day$^{-1}$) during the period October 2019 to October 2021. A gap in the graph indicates a missing data.

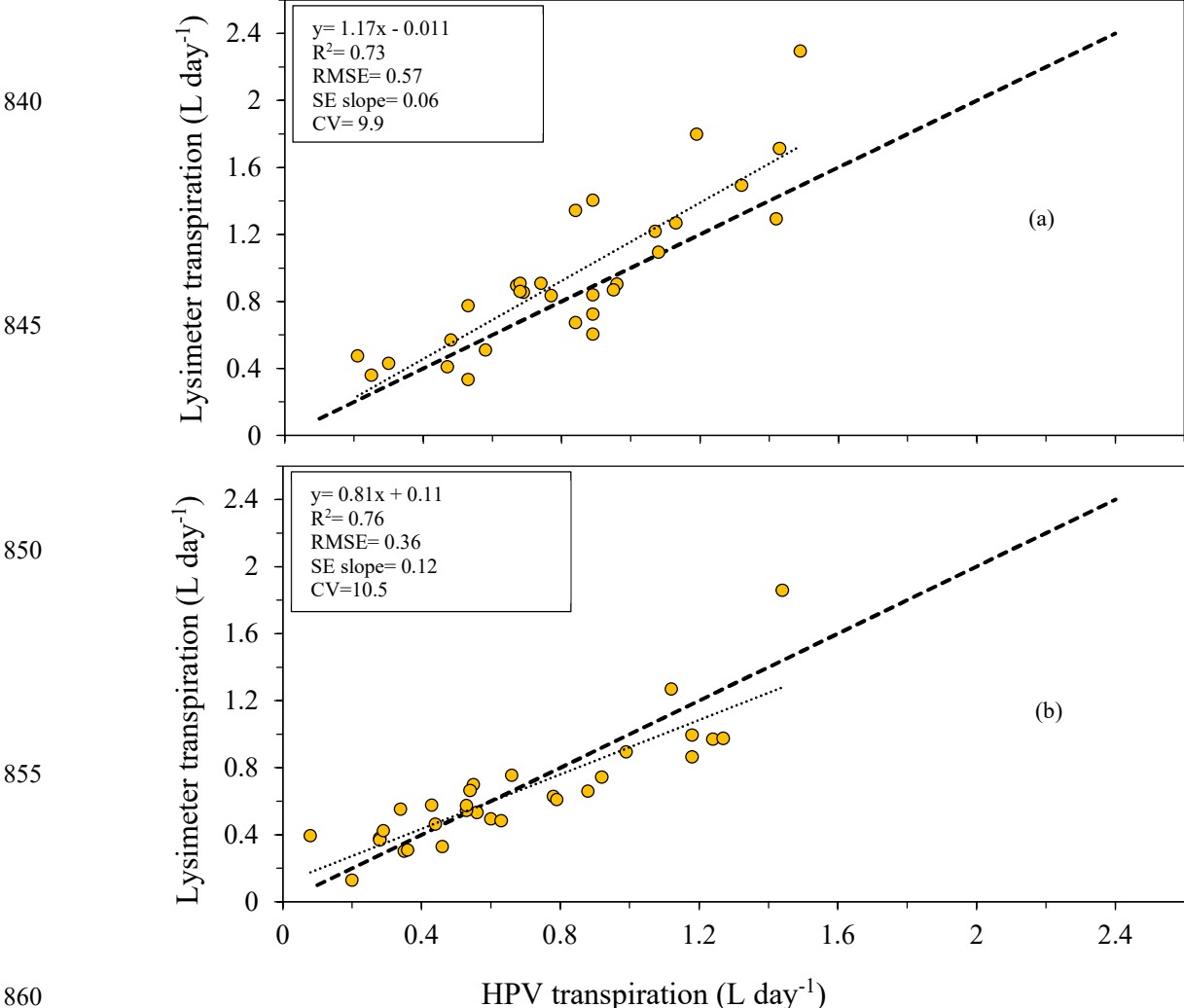

**Figure 5.** Relationship between daily transpiration measured using a heat ratio technique (HPV, L day$^{-1}$) and the transpiration measured using a lysimeter (through a change in mass, L day$^{-1}$) for (a) two-year-old *Eucalyptus grandis* x *Eucalyptus nitens* clonal hybrid and (b) three-year-old *Pinus elliottii*. The equation of the regression line, regression coefficient ($R^2$), root mean square error (RMSE), standard error of the regression slope (SE slope) and coefficient of variation (CV) for each species is presented. The dashed line is the 1:1 line.

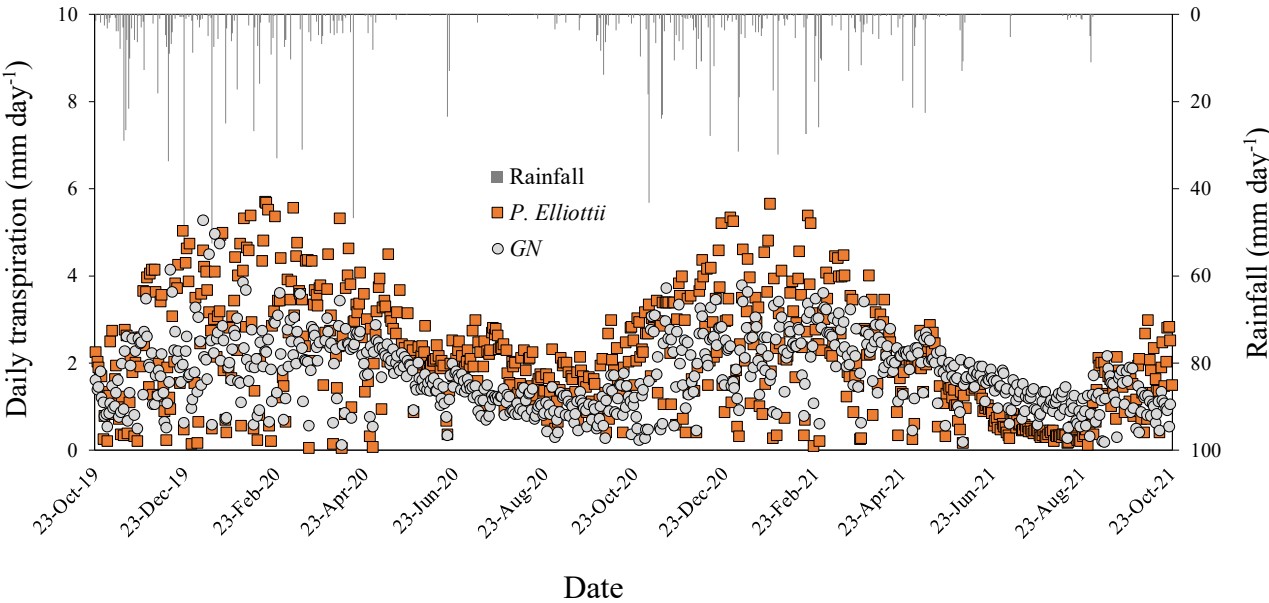


**Figure 6.** Mean daily transpiration ($T$, mm day$^{-1}$) and corresponding rainfall (mm day$^{-1}$) in an 8-year-old *E. grandis* x *E. nitens* clonal hybrid (*GN*) and 20-year-old *P. elliottii* trees for a duration October 2019 to October 2021. Each point is a mean of four trees.




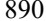

**Figure 7.** A daily tree transpiration (mm) of *P. elliottii* and *Eucalyptus grandis* x *E. nitens* clonal hybrid (*GN*) with corresponding daily rainfall (mm day⁻¹), daily vapour pressure deficit (kPa) and daily profile water content (PWC, mm per 0.6 m soil depth) over (a) a representative period in the summer season (24 October to 11 November 2020, the wet season) and (b) a representative period in the winter season (01 July to 19 July 2020, the dry season).


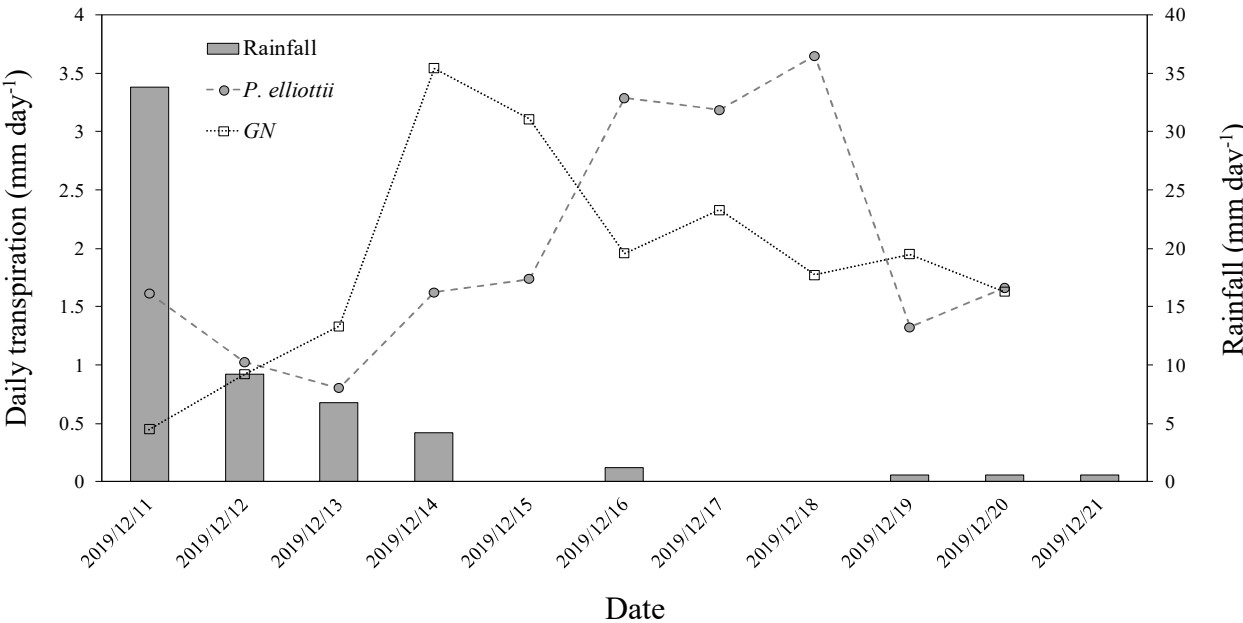


**Figure 8.** Ten-day daily transpiration (mm day⁻¹) for 20-year-old *P. elliottii* and 8-year-old *Eucalyptus grandis* x *Eucalyptus nitens* clonal hybrid (*GN*) with corresponding rainfall (mm day⁻¹) showing transpiration response by each specie to rainfall.




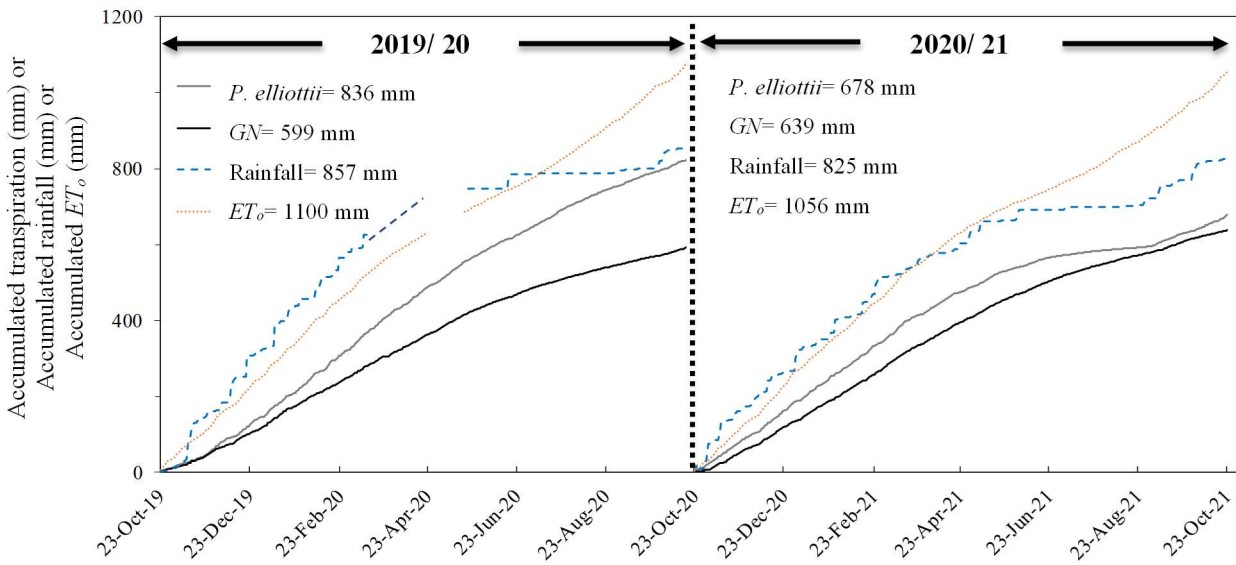


**Figure 9.** The accumulated transpiration (*T*, mm), rainfall (mm) and FAO reference evaporation ($ET_o$, mm) for 2019/ 20 hydrological year (Oct 2019 to Oct 2020) and 2020/ 21 hydrological year (Oct 2020 to Oct 2021).



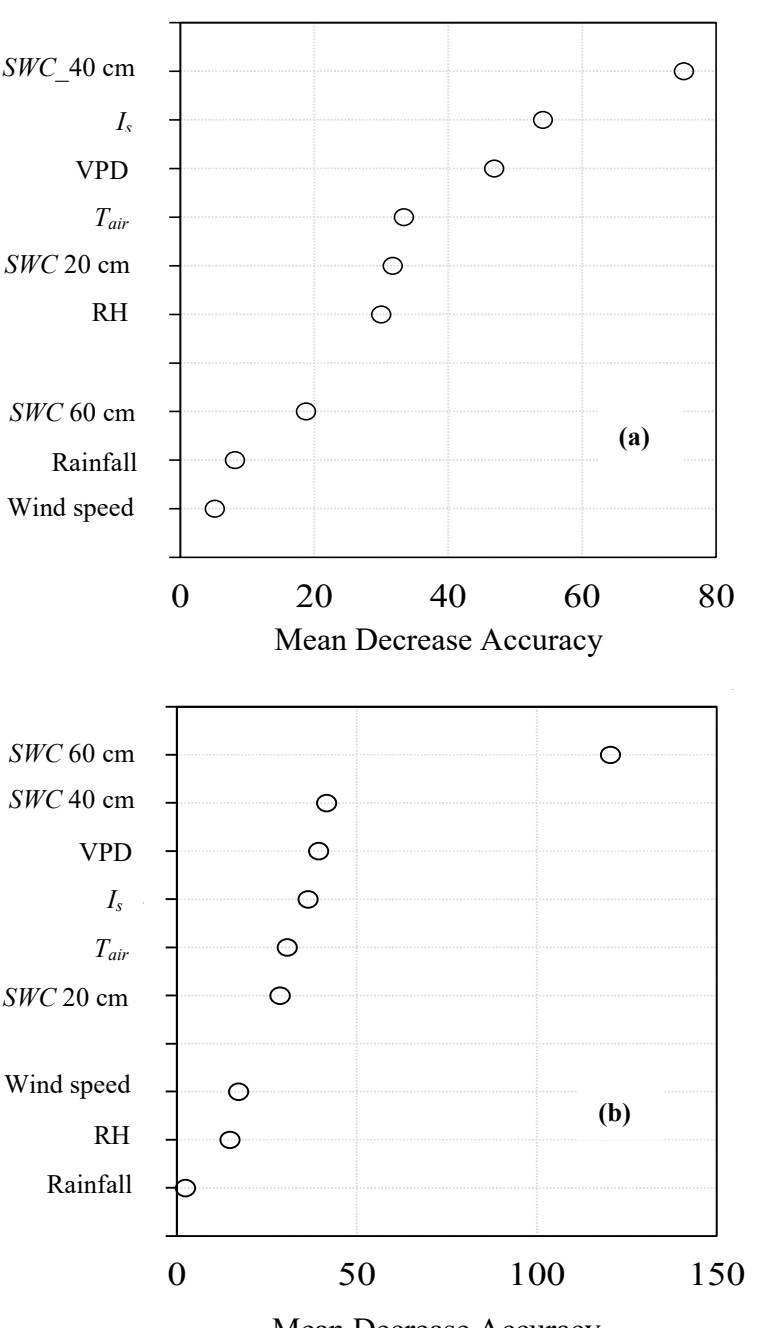

Figure 10. Variable importance plots of (a) *Eucalyptus grandis* x *E. nitens* (*GN*) and (b) *Pinus elliottii* from the random forest

regression model using solar radiation ($I_s$), vapor pressure deficit (VPD), air temperature ($T_{air}$), relative humidity (RH), wind

speed, rainfall and soil water content measured at 20 cm depth (*SWC* 20 cm), 40 cm depth (*SWC* 40 cm) and 60 cm depth

(*SWC* 60cm). Mean Decrease Accuracy is a measure of how much the model error increases when a particularly variable is randomly permuted.

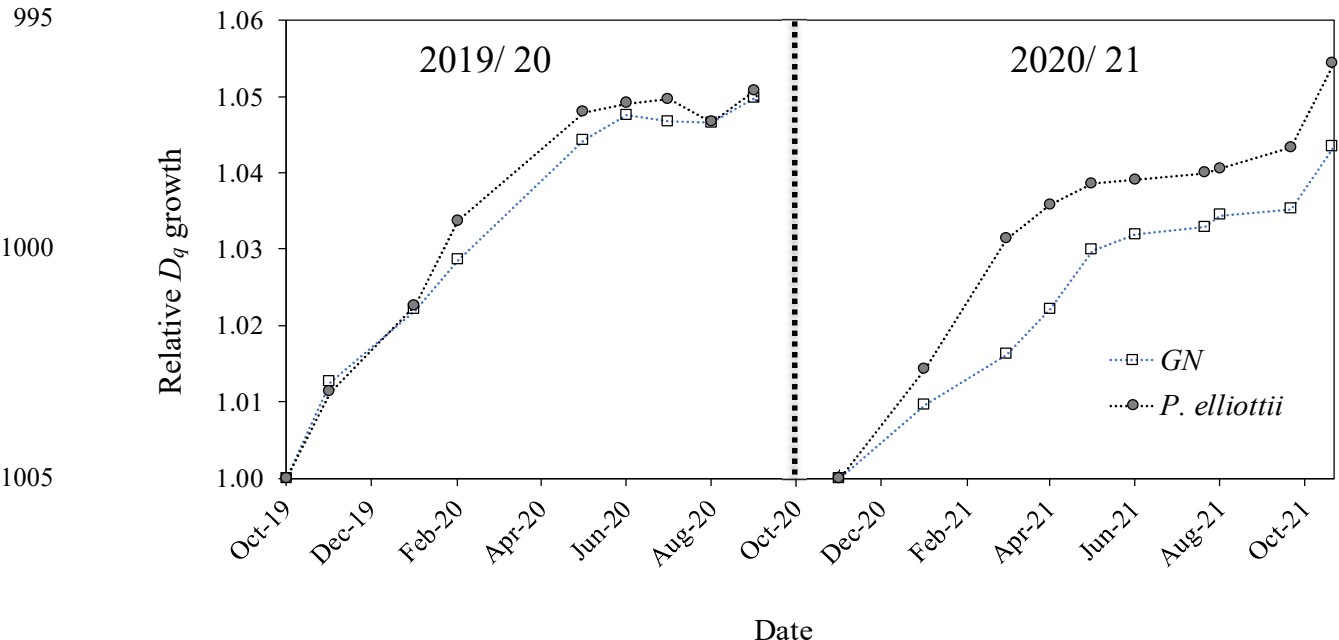

**Figure 11.** Relative quadratic mean diameters (*D_q*, normalised) measured using manual dendrometers bands for *Eucalyptus grandis* x *E. nitens* (*GN*) and *Pinus elliottii*. Measurements were conducted in two hydrological years, 2019/ 20 (October 2019 to October 2020) and 2020/ 21 (October 2020 to October 2021). Each point represents an average of 48 trees for each species.

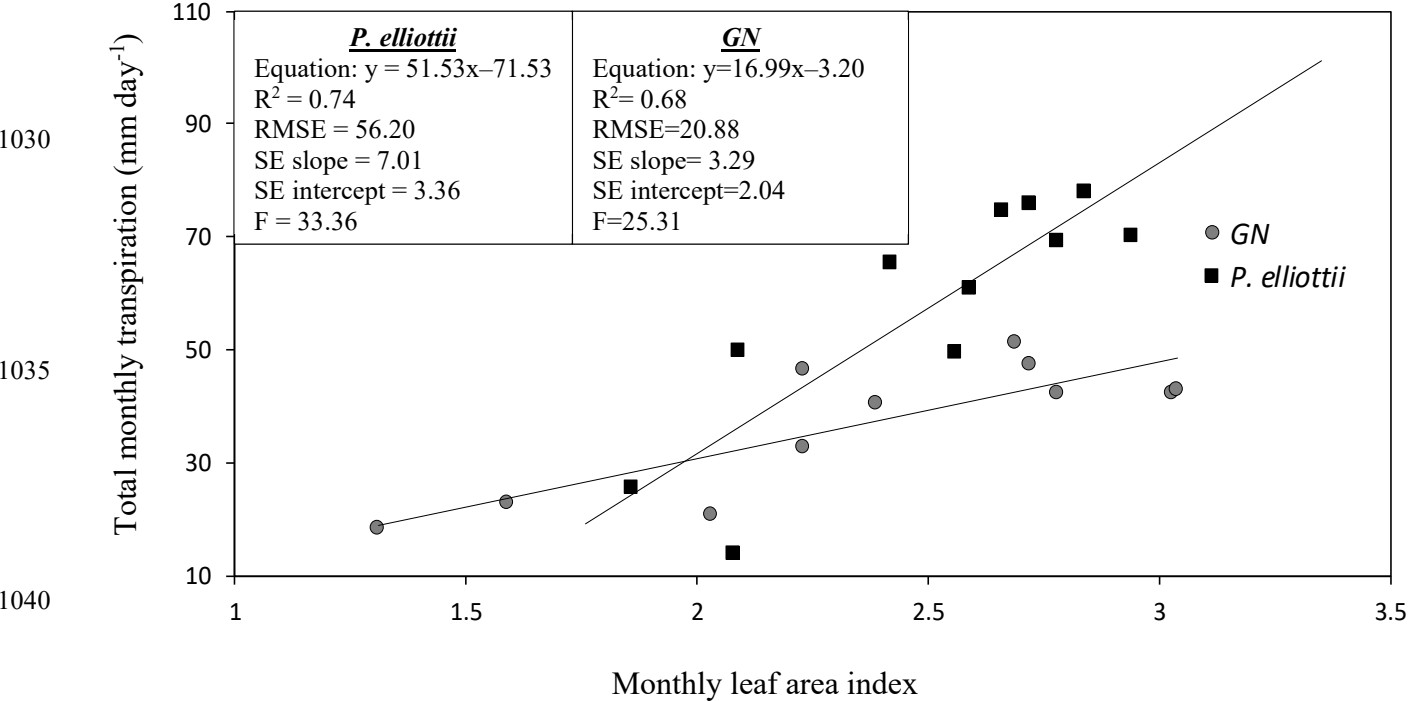

**Figure 12.** A linear relationship between total monthly transpiration ($T$, mm day$^{-1}$) and monthly measured leaf area index (LAI) for *Eucalyptus grandis* x *E. nitens* clonal hybrid (*GN*) and *Pinus elliottii*. The equation of the regression line, regression coefficient ($R^2$), root mean square error (RMSE), the standard error of the regression slope (SE slope), the standard error of the y- intercept (SE intercept) and the ratio of variance (F) for each species is presented.

 **Table 1.** The general characteristics of the two study sites at Mistley Canema. The abbreviations MAP and MAT denotes mean annual precipitation and mean annual temperature, respectively.

| | Study sites | |
|---|---|---|
| **Characteristics** | *P. elliottii* | *GN* |
| Lithology | Arenite | Arenite |
| Soil texture | Sandy loam | Sandy clay |
| Bulk density (g.cm$^3$) | 1.33 | 1.17 |
| Altitude | 884 | 976 |
| Climate | Warm temperate | Warm temperate |
| MAP (mm) | 800 – 1200 | 800 – 1200 |
| MAT (°C) | 17 | 17 |

**Table 2.** A general description of substrate (soil and geology) characteristics based on characterisation conducted at the Two
 Streams research catchment, which is adjacent to our study site (sourced from Clulow et al. 2014).

| **Horizons** | **Approximate depth (m)** |
|---|---|
| Orthic A | 0 – 0.25 m |
| Red apedal B | 0.26 – 13 m |
| Saprolite | 14 – 20 m |
| Grey fine-grained shale | 21 – 40 m |
| Grey fractured basement granite | 41 – 80 m |

**Table 3.** Detailed description of trees monitored on *Pinus elliottii* and *Eucalyptus grandis* x *E. nitens* clonal hybrid (*GN*) study
 sites.

| Trees | Overbark diameter (cm) | | Bark (cm) | | Sap-wood depth (cm) | | Probe depth under bark surface (cm) | |
|---|---|---|---|---|---|---|---|---|
| | *P. elliotti* | *GN* | *P. elliotti* | *GN* | *P. elliotti* | *GN* | *P. elliotti* | *GN* |
| Tree 1 | 10.7 | 10.5 | 2.2 | 0.7 | 4.88 | 2.55 | 1.0 | 1.0 |
| Tree 2 | 15.9 | 11.4 | 2.4 | 0.8 | 7.2 | 2.8 | 2.0 | 1.5 |
| Tree 3 | 18.2 | 12.5 | 2.4 | 0.8 | 8.3 | 3.0 | 3.0 | 2.5 |
| Tree 4 | 22.4 | 14.2 | 2.5 | 0.9 | 10.2 | 3.9 | 4.0 | 3.5 |