# Peer review of "Transpiration rates from a mature *Eucalyptus grandis* x *E. nitens* clonal hybrid and *Pinus elliottii* plantations near the Two Streams Research Catchment, South Africa"

_EGUsphere, 2022_

## Referee Comment (RC2)

[referee-annotated manuscript omitted]

---

## Author Response (AR1)

**Response to the reviewers**

The reviewers are sincerely thanked for their contribution to this body of work. Responses to all comments can be found below and those in the annotated documents have also been addressed and highlighted in yellow.

**Reviewer 1 (Miriam Coenders-Gerrits)**

With pleasure I read the manuscript of Kaptein et al. It deals with an important question whether the newly planted GN use more water then Pinus elliottii. The authors compared two sites and equiped several trees with sapflow sensors to quantify the transpiration. As the HPV-system has many limitations in case one wants to know the stand transpiration, the authors did an attempt to quantify the relation between the sapflow measurements and the transpiration rates determined via a lysimeter. My compliments for doing this!

Overall, I like the study very much. It is relevant, well written and easy to read. I only have the following comments, which can improve the manuscript:

1. you conclude that the transpiration is GN is lower than in Pinus due to the lack of water. Of course this is true: transpiration reduces as water becomes more limited (fig 5). However, I wonder if this very low matrix potential is not caused by the high transpiration rates of GN before the study period? So that the GN depleted already the soil? (feedback mechanism). **Authors agree with this possibility. A statement suggesting this possibility has been added in the manuscript line 277 to 282. The important aspect is perhaps that GN have the potential for a high-water use if the water is available but over the long-term, they depleted the soil water reserves and over the measurement period the water use was reduced as a result.**

2. The contribution of fog. In line 70 you say that fog precipitation is significant. How does this affect your results? How reliable are your precipitation observations? And how would this affect your interception results? **We have modified the statement to read "The area experiences mist which could significantly contribute to overall precipitation through streamflow or canopy interception" (line 66 to 67). The rainfall has been measured accurately with a Texas Instruments research grade raingauge and with a second station nearby to collaborate our results. The accuracy of the rainfall results would impact the interception because the interception equation is based on the rainfall.**

3. Units/dimensions: I think the authors should do a carefull check on the units. In my view transpiration is a flux, thus having a time dimension. Therefore, most of the numbers given in the paper should have the unit mm/day or mm/month or mm/year (instead of mm). Some quick examples: L19, 151 (liter), 183 (mm/year), 213, 214, 215, 242, 265, 285 (mm/year), fig 3d (mm/day) and caption, fig 5 (rainfall mm/day), fig 5 (T=mm/day), fig 6 (rainfall mm/day), fig 8 (rainfall mm/day) and caption, fig 11 (mm/month) plus caption. **Agreed thank you. Completed throughout the manuscript.**

4. Equations 3 +4: I would make this liniear. No need (and reason) for polynomial. **Equations changed from polynomial to linear equation (line 219 to 225).**

5. Section 4.3: I would keep this section for qualitative as you did not measure interception and soil evaporation. Especially, since you have fog the interception could be very high. **Authors have removed this section and focused on the measured tree transpiration data.**

6. L187: unit of Is should be MJ/m2/d. Also check fig 3b. **Amended throughout the manuscript as suggested.**

7. I would recommend to write all parameters in the text in italic and make use of subscript (e.g., Tair => $T_{air}$): **Completed throughout the manuscript.**

**Reviewer 2 (David Scott)**

An interesting paper, representing a lot of difficult work, on a topic that receives little detailed investigation, despite its significance. Thus I consider the work worthwhile and worthy of publication, provided that several points are attended to in revision. Whether these represent minor or major revision is a matter of opinion.

1. the comparison between eucalypts and pine is relevant, but there is not enough hard data of make a definitive determination of their relative effects. I think that the title and abstract should soften the emphasis on this comparative water use. **Agreed and thank you for the useful suggestion. The word "comparative" in the article title was removed and the words "comparison" or "comparative" were removed from the abstract.**

2. Interesting data is presented on the comparative transpiration, but the other elements of total water use are poorly quantifiable, so the final comparison of ET by the two species is little better than speculative. **Agreed. Authors have removed the section with the comparison in ET between the two crops and focused on the measured tree transpiration.**

3. Specifically, the crops are very different ages and thus a direct comparison is not convincing (at least not without a lot more convincing information on why a comparison is reasonable. *Eucalyptus* **trees in South Africa are grown over 12-year rotation, while Pine trees are grown over a 25-year rotation. So, at 8 years for** *Eucalyptus grandis* **x** *E. nitens* **and 20 years for** *P. elliottii***, they were as similar stages in their growth cycles. This has now been highlighted in the manuscript (line 55 to 57).**

4. Secondly, the soil water component is poorly defined, and the soil & interception figures are estimated from such general sources (of dubious applicability) that the errors on those estimates make the overall estimates speculative. I think these are serious flaws, but I think the information on direct measurements in the paper are still important and useful. But the authors should not attempt to close the water balance except in a broad and speculative way. **The soil water content section has been improved (Line 129 to 142, line 198 to line 216). The sections that estimated ET has been removed and authors focused on direct measurements (transpiration) as suggested.**

2. My annotated version of the manuscript contains several parts where the language or expression is lacking and can be improved. **Thank you for these notes. Language has been improved throughout the manuscript as suggested.**

3. Section 2.5 on the soil water content left me puzzled, and I think that it and the later references to the soil water store, are wrong. Soil wetness is presumably measured with some sort of TDR instrument, that give a volumetric wetness at the 3 depths (this should be stated more clearly in the text - I don't know what a CS616 sensor is). **Clarity on the CS616 sensor has been provided: line 131 to 133 including a description of the profile soil water content calculation (Line 137 to 142 and line 198 to line 216) .**

(a) These volumetric wetness figures could be used to estimate a depth of water in the profile, which is a measure of the absolute amount of water available to the trees (PAW). It doesn't make sense to me to go through some sort of model to estimate the approximate soil water potentials that these wetness figures would relate to (section 3.2). If the objective is to consider plant available water, then it is best to use volumetric wetness, and convert these to an estimate of a depth of water in the soil profile at a each particular point in time. **Soil water potential was calculated through measured retention curves, but we agree that using volumetric wetness is most suitable and the soil water potential results have been removed. The measured soil water content was used to estimate the soil profile water content (line 129 to line 142) and results presented in Figure 4 and line 198 to 216.**

(b) With the deepest probe at 60 cm (properly 0.6 m in SI units), there is a large uncertainty about how much water is still available to the trees at depths below say 0.75 m? The estimates of soil water are therefore incomplete, and only give an indication of the real situation. The information on the response of the trees to increase water availability immediately after rain, or in the dry season, are interesting and useful, but it needs to be acknowledged that the trees may have access to water in parts of the profile that have not been monitored. {Consider Peter Dye's measurements of eucalypts under stress - 1996, Tree Physiology, 16: 233-238, which showed water was being used to 8 m below the surface and beyond}. **Agreed. A statement acknowledging that trees may have accessed soil water beyond 0.6 m was incorporated (line 209 to 216).**

(c) Related to the point in (b), more should be said about the soil profiles, and this should include materials below the conventional soil profile and include the substrate beneath, from which water may be drawn. Is it hard rock, or is it decomposing and permeable (as well as accessible to tree roots?) **Beneath the soil profile, there is weathered, unconsolidated material that is accessible to tree roots. This information has been incorporated (Line 73 to 76 and Table 2).**

---

## Referee Report (RR1)

**Notes on Kaptein et al. (revised) "Water use by . . ."**

The title says "water use" which is misleading.  The authors hoped that this might be what could be determined, but in reality this paper is about comparative **transpiration** (rates) in pine & eucalypts plantations over two years.

Abstract: pine plantations are not a specie;  "dominant type" of plantation perhaps.

There is no singular (specie) for species: it is one species and two species.

2019/20 for a single growing year?  I think this or 2019_20 to show a hydrological or growing season is easier to understand.

Annoying overuse of abbreviations, particularly in areas such as discussion. For example, GN T rather than transpiration from eucalypts.  I suggest the authors specify the trees, but that they can then simply refer to the two crops as pines and eucalypts.

**Profile soil wetness**

$PWC_{\square\square} = (SWC_{\square\square} \times 0.2) + (SWC_{\square\square} \times 0.2) + (SWC_{\square\square} \times 0.2)$

*If SWC is volumetric wetness (%) {this should be specified in text} then*
*PWC would return a depth of water in m in the top 0.6 m.  Text talks about mm of water use, so change formula to give mm of water in the 0.6 m of profile.*

*Section 3.2:*

*Figures 4 & 6.  Nice figures, but too compressed to facilitate understanding of soil water response to rainfall.*
*I like Figure 7 as it explores the detail available when one looks at shorter intervals.  Perhaps figures 4 & 6 could be broken up to a greater extent to illustrate more of the detailed information available on T & other potentially driving variables (over short periods of time).  To my mind, the real value of the study is in the information contained in the detailed transpiration data and the associated environmental variables.  The current figures 4 & 6 are fine, but only if accompanied by others that make more of this information.  How about a short interval (2 weeks to a month) in late dry season when VPD is high but T is perhaps low; another in peak summer conditions when water supply is likely not limited; etc.*

Figure 8: accumulated data: correct units are now mm (not mm/day as you've added the mm from each day to the running total)

TABLE 2.
Is it the same at both sites?  seems unlikely so indicate that this is a general picture based on . . .
Most soil scientists would not be looking (or including) material below 1 m (or the well-developed soil profile, so I suggest you call this the substrate (soil & geology)
Also, Orthic-A, by definition*, is unlikely to be more than 10 - 12 CM deep (not to mention m).
     An * orthic-A is a poorly developed A-horizon of limited thickness and low organic matter content.

Table 3:

Probe depth under bark for pines: 1, 2, 3 & 4 cm seems a bit strange (no decimals, and an orderly progression)??

Table 4:
This looks like nonsense.  I presumed from the text that various **independent variables** were tested as predictors of the **dependent variable** which is Transpiration.  If these are tested individually it would seem to me to be a simple correlation (not an Anova) – so please clarify.  Are you reporting the result of a multiple regression, or are these p (probabilities) associated with correlation statistics?
If these are derived from an Anova, then I think the model you're presenting needs fuller explanation in the text (Methods).  *{this was my first impression, but having read section 3.5 again, I suggest that the comments below cover my concerns more accurately}*

Section 3.5.
Multi-variable regression of the sort reported here (and Table 4) is interesting, but can also be somewhat misleading because of auto-correlation between the independent variables (radiation load, RH and VPD for instance will not be fully independent of each other on a particular day). So, to look at which variable best "explains" T, one should do a correlation matrix with each variable on its own.  Now you might see several strong correlations, which might be more informative.  It will also illustrate correlations between the various "controlling variables".  The multiple regression is needed (useful) only when one desires to predict T using the available information.  I suggest that in this study it is more useful to understand how T correlates to various environmental factors. The explanation given in this section (3.5) is therefore flawed as it is probably misleading.  Dye has shown that VPD is a robust predictor of T, but if you only look at the results of a multiple regression analysis, VPD may appear as of little use, but only because the same information is contained in other variables, which in this particular instance do a better job of prediction. One is thus obscuring the picture (of what drives transpiration) that all this hard work is trying to clarify*. [Then one may add a multiple regression IF it is thought that the predictive equation is a necessary tool to derive from this work.]*

In section 4.1 and earlier, "*With our GN trees 8-years-old (full rotation is 12 years)"* the authors are treating a commercial forestry rotation as though it indicated some real 'life stage' in the trees.  This is misleading as the forestry company simply cuts the trees at an age aimed to maximize profits.  Therefore, I suggest one rely on tree age as a more reliable indicator of life stage, though it is relevant to state that trees are large enough within X years to be harvested for pulping.  It is also relevant that eucalypts clearly grow very quickly in the right conditions and can have high LAI and transpiration rates when still young (I suggest you refer to earlier SA work by Dye & colleagues in making this point).

Lines 301 – 305, my impression: VPD no longer influenced T in winter (or later on a sunny day) because there's a shortage of accessible water in the soil (so atmospheric demand may be high but the tree is unable to respond)?

The only reference to the substantial body of paired catchment studies in South Africa is Scott & Lesch, which is not really the most significant of a number of important papers.  There is clear evidence from these catchment studies that eucalypts have an early and large impact on streamflow (high ET) relative to pine plantings.  I think that this point should be made and it should be stressed that the results in this study are not in line with the general & longer term picture shown from many years of catchment studies in SA.

As for the comparison between pines & eucalypts generally: native eucalypts in Australia are not as vigorous as the exotic pines, so there the comparison is different from in South Africa and So. America where both types are exotic (and the eucalypts are freed from constraints of local biological controls).

Lines 340 -345.  Well summed up.  Again, reference should be made to the SA catchment studies that are the most relevant comparison.  The subject of timber plantation water use and effects on water supply, is a complex one, but to ignore part of the information that is available doesn't do the authors any credit.  The awkward part of studies of transpiration, stomatal conductance, interception, soil water content, etc., is that the integrated effect of all the components of the hydrological process is difficult to obtain, and one is left with an interesting, informative picture of a part of the whole process (in other words a partial picture).  This is why long-term catchment experiments are so useful.  A catchment study integrates all the processes, and provides an integrated answer over a long period of time.  This point is well illustrated with the current study.  We have to speculate about what might have happened with evapo-transpiration and soil water stores in the years before the measurements began, and there is uncertainty about how deeply the trees are extracting water from the profile.  Everyone cannot do catchment experiments (for reasons of time & money) but it makes no sense to ignore the results from those that are available.  The authors should look first to the most relevant comparison which is the SA studies in the summer rainfall region, then the pine studies in Jonkershoek.  The truth is that in **no** catchment study with eucalypts in SA (Mokobulaan, Westfalia or Ntabamhlope, covering a considerable MAP range) has the stream not dried up. This is the clearest, fully integrated picture of the hydrology of eucalypt plantations in SA conditions that exists.  To ignore this fact in a comparison of eucalypts and pines, and talk about the Australian results or a partial picture from Chile, to imply that eucalypts may not have a large impact, is poor science and misleading (however much the SA timber companies may like to hear it).  So, I think that this point (i.e. what the catchment studies tell us) needs to lead the discussions.

To my mind, the results obtained in this study and those of catchment studies or Peter Dye, are not in contradiction, except superficially.  However, it requires that one consider the longer term picture (multiple years of rainfall & evaporation, with carry over effects from year to year. The catchment studies show that pines develop a canopy more slowly and their effect on ET (& hence streamflow) takes longer to be apparent.  In some years the plantations use more water than arrives as precipitation, but water must be coming out of deeper stores to supply this use. Hence, what one measures in a single year is simply a snapshot, and cannot be expected to provide a complete answer.  Limited transpiration in the case of eucalypts might be explained by current water availability (as suggested by the authors in the discussion) and does not imply that eucalypts will generally have a lower transpiration rate than pines.

[revised manuscript text omitted]

---

## Author Response (AR2)

**Response to the reviewers**

Reviewer#1 and 2 are thanks for their valuable contribution to the manuscript. Responses to all comments can be found below and those in the annotated documents have also been addressed and highlighted using track changes.

**Reviewer # 1**

- minor correction:T in abstract should be mm/y. **Thank you for this correction. Corrected (line 18).**
- same in section 4.2 **Reviewer #1 is thanks for these corrections. Correction completed (line 353 to 354).**

**Reviewer #2**

- The title says "water use" which is misleading. The authors hoped that this might be what could be determined, but in reality this paper is about comparative transpiration (rates) in pine & eucalypts plantations over two years. **Thank you for this comment. Previous comments from reviewer #2 stated "the comparison between eucalypts and pine is relevant, but there is not enough hard data to make a definitive determination of their relative effects. I think that the title and abstract should soften the emphasis on this comparative water use. Authors responded "Agreed and thank you for the useful suggestion. The word "comparative" in the article title was removed and the words "comparison" or "comparative" were removed from the abstract". Based on this suggestion, authors revised the title to "A comparison of water-use by fast growing *E. grandis* x *E. nitens* clonal hybrid and *P. elliottii* near the Two Streams research catchment, South Africa".**
- Abstract: pine plantations are not a specie; "dominant type" of plantation perhaps. There is no singular (specie) for species: it is one species and two species. **Reviewer is thanks for this correction. A word "specie" has been changed throughout the manuscript to "species".**
- 2019/20 for a single growing year? I think this or 2019_20 to show a hydrological or growing season is easier to understand. **Thank you for this suggestion. The hydrological year "2019' 20" has been changed to "2019/ 20" while "2020' 21" has been changed to "2020/ 21" throughout the manuscript.**
- Annoying overuse of abbreviations, particularly in areas such as discussion. For example, GN T rather than transpiration from eucalypts. I suggest the authors specify the trees, but that they can then simply refer to the two crops as pines and eucalypts. **Reviewer is thanked for this suggestion. Abbreviations have been reduced in the manuscript by 1) not abbreviating the word "transpiration" and writing it in full 2) writing the abbreviation "LAI" in full as leaf area index.**
- Profile soil wetness $PPPPPP\diamond.\diamond = (SSSSSS\diamond.\diamond \times 0.2) + (SSSSSS\diamond.\diamond \times 0.2) + (SSSSSS\diamond.\diamond \times 0.2)$ If SWC is volumetric wetness (%) {this should be specified in text} then PWC would return a depth of water in m in the top 0.6 m. Text talks about mm of water use, so change formula to give mm of water in the 0.6 m of profile. **Thank you for the correction. The formula has been corrected to reflect mm of water per 0.6 m of the soil profile (Equation 2).**

- Section 3.2: Figures 4 & 6. Nice figures, but too compressed to facilitate understanding of soil water response to rainfall. I like Figure 7 as it explores the detail available when one looks at shorter intervals. Perhaps figures 4 & 6 could be broken up to a greater extent to illustrate more of the detailed information available on T & other potentially driving variables (over short periods of time). To my mind, the real value of the study is in the information contained in the detailed transpiration data and the associated environmental variables. The current figures 4 & 6 are fine, but only if accompanied by others that make more of this information. How about a short interval (2 weeks to a month) in late dry season when VPD is high but T is perhaps low; another in peak summer conditions when water supply is likely not limited; etc. **Reviewer #2 is thanked for this suggestion. Figures showing short term picture of tree transpiration, profile water content, tree transpiration and VPD have been incorporated in the manuscript (Figure 7a and 7b).**

- Figure 8: accumulated data: correct units are now mm (not mm/day as you've added the mm from each day to the running total). **Thank you for this correction. The figure has been corrected (Figure 8).**

- TABLE 2. Is it the same at both sites? seems unlikely so indicate that this is a general picture based on . . . **Thank you. A statement indicating that Table 2 is a general picture has been added (line 84 to 85 and the caption on Table 2).**

- Most soil scientists would not be looking (or including) material below 1 m (or the well-developed soil profile, so I suggest you call this the substrate (soil & geology) Also, Orthic-A, by definition*, is unlikely to be more than 10 - 12 CM deep (not to mention m). An * orthic-A is a poorly developed A-horizon of limited thickness and low organic matter content. **Thank you for this correction, Authors agree, Orthic-A is a very shallow layer reported to be as deep as 30 cm. Authors corrected the Orthic A depth in Table 2. Furthermore, a correction was done in the caption of Table 2 to refer to material below 1 m as substrate (soil and geology Table 2).**

- Table 3: Probe depth under bark for pines: 1, 2, 3 & 4 cm seems a bit strange (no decimals, and an orderly progression)?? **We thank reviewer for this comment. The probe insertion for pine is correct, the first probe was installed at 10 mm under bark and other probes at 10 mm increments thereafter, guided by the core sample that was collected before tree instrumentation.**

- Table 4: This looks like nonsense. I presumed from the text that various independent variables were tested as predictors of the dependent variable which is Transpiration. If these are tested individually it would seem to me to be a simple correlation (not an Anova) – so please clarify. Are you reporting the result of a multiple regression, or are these p (probabilities) associated with correlation statistics? If these are derived from an Anova, then I think the model you're presenting needs fuller explanation in the text (Methods). {this was my first impression, but having read section 3.5 again, I suggest that the comments below cover my concerns more accurately}. Section 3.5. Multi-variable regression of the sort reported here (and Table 4) is interesting, but can also be somewhat misleading because of auto-correlation between the independent variables (radiation load, RH and VPD for instance will not be fully independent of each other on a particular day). So, to look at which variable best "explains" T, one should do a correlation matrix with each variable on its own. Now you might see several strong correlations, which might be more informative. It will also illustrate correlations between the various "controlling variables". The multiple regression is needed (useful) only when one desires to predict T

using the available information. I suggest that in this study it is more useful to understand how T correlates to various environmental factors. The explanation given in this section (3.5) is therefore flawed as it is probably misleading. Dye has shown that VPD is a robust predictor of T, but if you only look at the results of a multiple regression analysis, VPD may appear as of little use, but only because the same information is contained in other variables, which in this particular instance do a better job of prediction. One is thus obscuring the picture (of what drives transpiration) that all this hard work is trying to clarify. [Then one may add a multiple regression IF it is thought that the predictive equation is a necessary tool to derive from this work.]. **Authors thanks Reviewer #2 for these insightful comments and agree. The idea of conducting a multiple regression was to identify climatic variables that influence transpiration with an intention to ultimately use the most responsive variables as predictors of transpiration in future modelling studies as climatic variables are easy to measure compared to in situ transpiration method. This is why multiple regression analysis was conducted. Authors decided to use a different approach, by using Random Forest regression model to correlate tree transpiration with climatic variables where the contribution of each climatic variable to the model was investigated. Complete data re-analysis of relationship between transpiration and each climatic variable was conducted and results presented (line 24 to 27, line 190 to 207, line 275 to 294, line 363 to 380, line 407 to 409). Therefore, Table 4 was removed from the manuscript.**

- In section 4.1 and earlier, "With our GN trees 8-years-old (full rotation is 12 years)" the authors are treating a commercial forestry rotation as though it indicated some real 'life stage' in the trees. This is misleading as the forestry company simply cuts the trees at an age aimed to maximize profits. Therefore, I suggest one rely on tree age as a more reliable indicator of life stage, though it is relevant to state that trees are large enough within X years to be harvested for pulping. It is also relevant that eucalypts clearly grow very quickly in the right conditions and can have high LAI and transpiration rates when still young (I suggest you refer to earlier SA work by Dye & colleagues in making this point). **Reviewer is thanked for these comments. Authors agree that the length of the rotation is influenced by several factors such as climate, type of soil, species etc, but generally, a eucalypt rotation grown for pulp ranges from 10 to 12 years in South Africa. Most commercial forest plantation producers will less likely harvest Eucalyptus before the age of approximately 10 years, even in high productive sites inorder to maximise yields. In less productive sites, Eucalyptus rotation is usually around 12 years increasing to 15 years in certain cases. Authors, used an average rotation value of 10 years as a benchmark in this study, subject to change depending on the forest producer. A statement suggesting this has been highlighted in the manuscript (line 61 to 67). There is a general consensus that in the early stages of Eucalyptus growth, transpiration rates are high accompanied by high leaf area index, reaching a peak in the middle of the rotation, thereafter, declining as the stand matures.**

- Lines 301 – 305, my impression: VPD no longer influenced T in winter (or later on a sunny day) because there's a shortage of accessible water in the soil (so atmospheric demand may be high but the tree is unable to respond)? **Thanks for this suggestion. Authors agree and this statement has been incorporated in the manuscript (line 350 to 351).**

- The only reference to the substantial body of paired catchment studies in South Africa is Scott & Lesch, which is not really the most significant of a number of important papers. There is clear evidence from these catchment studies that eucalypts have an early and large impact on streamflow (high ET) relative to pine plantings. I think that this point should be made and it should be stressed that the results in this study are not in line with the general & longer term picture shown from many years of catchment studies in SA. As for the comparison between pines & eucalypts generally: native eucalypts in Australia are not as vigorous as the exotic pines, so there the comparison is different from in South Africa and So. America where both types are exotic (and the eucalypts are freed from constraints of local biological controls). Lines 340 -345. Well summed up. Again, reference should be made to the SA catchment studies that are the most relevant comparison. The subject of timber plantation water use and effects on water supply, is a complex one, but to ignore part of the information that is available doesn't do the authors any credit. The awkward part of studies of transpiration, stomatal conductance, interception, soil water content, etc., is that the integrated effect of all the components of the hydrological process is difficult to obtain, and one is left with an interesting, informative picture of a part of the whole process (in other words a partial picture). This is why long-term catchment experiments are so useful. A catchment study integrates all the processes, and provides an integrated answer over a long period of time. This point is well illustrated with the current study. We have to speculate about what might have happened with evapo-transpiration and soil water stores in the years before the measurements began, and there is uncertainty about how deeply the trees are extracting water from the profile. Everyone cannot do catchment experiments (for reasons of time & money) but it makes no sense to ignore the results from those that are available. The authors should look first to the most relevant comparison which is the SA studies in the summer rainfall region, then the pine studies in Jonkershoek. The truth is that in no catchment study with eucalypts in SA (Mokobulaan, Westfalia or Ntabamhlope, covering a considerable MAP range) has the stream not dried up. This is the clearest, fully integrated picture of the hydrology of eucalypt plantations in SA conditions that exists. To ignore this fact in a comparison of eucalypts and pines, and talk about the Australian results or a partial picture from Chile, to imply that eucalypts may not have a large impact, is poor science and misleading (however much the SA timber companies may like to hear it). So, I think that this point (i.e. what the catchment studies tell us) needs to lead the discussions. To my mind, the results obtained in this study and those of catchment studies or Peter Dye, are not in contradiction, except superficially. However, it requires that one consider the longer term picture (multiple years of rainfall & evaporation, with carry over effects from year to year. The catchment studies show that pines develop a canopy more slowly and their effect on ET (& hence streamflow) takes longer to be apparent. In some years the plantations use more water than arrives as precipitation, but water must be coming out of deeper stores to supply this use. Hence, what one measures in a single year is simply a snapshot, and cannot be expected to provide a complete answer. Limited transpiration in the case of eucalypts might be explained by current water availability (as suggested by the authors in the discussion) and does not imply that eucalypts will generally have a lower transpiration rate than pine. **Reviewer #2 is thanked for such insightful suggestion. Authors agree, quantifying the impact of eucalypts and pine on water resources requires long term measurements. Even when long term measurements are conducted**

on a specific site, problems associated with climate variability is a challenge, making extrapolating results to other sites difficult. Long term catchment studies have been incorporated in the manuscript and a warning has been incorporated suggesting that these results can not be extrapolated to other areas due to differences in climate (line 401 to 431, line 462 and line 468 to 469).

- Weasel sentence: not clear and not convincing. **Thank you for this comment. The sentence has been revised and more explanation provided (line 61 to 67).**

- Shale (Ecca) on its own is of little value to a broader audience. **Thank you for this comment. Authors agree and it is indicated in the document that soils are shales that belong to Ecca group (line 75).**

- Unlikely. Would prefer a generic description of the climate here: Koppen class and broad description of conditions. **The reviewer is thanks for this suggestion. The Koppen-Geiger climate classification has been included in the manuscript (line 77 to 79).**

- Prior to canopy closure. **Thanks for this suggestion. This has been incorporated (line 91).**

- Generalisation: simply say that xylem was deeper. **Thank you. Corrected (line 113).**

- Sapwood. **Thank you. Incorporated (line 139).**

- Potential evapotranspiration? If so, say that. **Thank you. Corrected as suggested (line 221**).

- After. **Thank you. Corrected (line 236).**

- Say how much (mm/day for example) not just that it was significantly more less. **Thanks for this suggestion. Line 260 to 262 indicates the mean transpiration values.**

- Simply 60 mm/0.6 m (not per day). **Thanks. Corrected (line 271).**

- Although. **Thank you. Corrected (line 272).**

- T was most responsive to. **Thank you so much for this suggestion. This section has been completely revised (line 295 to 303).**

- What does this actually mean? I suggest it is included in the statement "there were stat differences in the regressions". **Thanks for this comment. This suggests that *Eucalyptus* regression line fits the data better than *P. elliottii*, therefore has more precise prediction of transpiration than *P. elliottii*. A statement indicating this has been included in the manuscript (line 314 to 315). A statement indicating there were statistical differences in the regression line has been included on line 310.**

- My thought: VPD no longer influenced T in winter because there was a shortage of accessible water in the soil (so atmospheric demand may be high but the tree is unable to respond). **Thanks for this suggestion. Authors agree and this statement has been incorporated in the manuscript (line 349 to 351).**

- Rather, "Other studies on pines…" **Thanks for the correction. Corrected (line 356).**

- Dye (1996) shows a similar persistence of T despite falling water supply. **Thank you. A finding by Dye 1996 have been incorporated (line 360 to 363).**

- Over a full rotation, let alone a number of rotations. **Thank you for this correction. This section has been revised (line 401 to 431).**

- Good summary, with right cautions. **Thank you.**

- Figure 1: Nice image but caption could be more friendly: explain symbols, and indicate what colour indicate (pine eucalypts and cane). **Thanks for this comment. The markers have been improved in the Figure and more explanation has been provided on the caption (Figure 1).**

- Figure 4: Figure may be more useful if split into 3 growing seasons. Resolutions is so cramped here that only general impressions are possible. **Thank you for this suggestion. A short-term picture of figure 4, showing a relationship between profile water content, tree transpiration, vapour pressure deficit and rainfall is shown in Figure 7a and 7b.**

- Figure 4: Profile water content is an absolute depth of water (per depth of soil: mm/ 0.6 m). It is not a rate (mm /day). **Thanks for the correction. Corrected (Figure 4).**

- Figure 7: Mean daily T plotted for a 10-day period in December 2019. **Thank you for the correction. Incorporated as suggested (Figure 7a and b).**

- Once the data is accumulated, the units become, simply, mm. **Thanks for the correction. Units corrected in Figure 8.**

- Figure 9: Units for Fig 9 are presumably mm? **Thank You. The relative quadratic mean diameter is a ratio between initially measured tree diameter and subsequent measurements thereafter, therefore it is unitless.**

---

## Author Response (AR3)

Authors would like to thank the editor and reviewers for their contributions to this paper, which have significantly added to the content of the paper.

- Many of the line numbers referred to in your response do not match those in version four of the manuscript, so it is sometimes difficult to assess your responses adequately. **Authors would like to sincerely apologize for mismatching line numbers with responses.**

- I do not follow the discussion over the title and I don't think that what you have suggested actually reflects the suggestions of the reviewer. Please consider something like "Transpiration rates of fast-growing Eucalyptus grandis x E. nitens clonal hybrid and Pinus elliottii near the Two Streams Research Catchment, South Africa". **Thank you for this suggestion and we agree. The title has been revised to "Transpiration rates from a mature *Eucalyptus grandis* x *E. nitens* clonal hybrid and *Pinus elliottii* near the Two Streams Research catchment, South Africa".**

- Your response around the age of the trees to reviewer 2 is inadequate. You state that generally a eucalyptus rotation grown for pulp ranges from 10-12 years in South Africa. However, this is not true - although it may have been in the past. See for example https://www.forestrysouthafrica.co.za/wp-content/uploads/2016/06/tree-rotations-1.pdf which shows that 7-10 years is the typical drawing length for eucalyptus. As the reviewer points out they will harvest the trees when the growth rate is optimal as when it starts to decline there is a loss of profit and so the trees are harvested. That drop in LAI is not so severe then - this is well explained in Gush et al., 2002. **The Editor is thanked for this correction and references. The section at the end of the introduction (line 60 to 64) has been updated to reflect the information provided.**

- The response to the long-term catchment recommendations of the reviewers is not that clear and I really don't think that you conclusions i.e. .."that, in contrast to common misperception, 1) P. elliottii can use more water than GN (depending on soil water stress)" is valid as 1) Almost all other literature shows that over a long rotation, eucalyptus does use more than pine, so it is not a misperception and 2) your study only takes place over two years and you showed higher water use by P. elliottii in only one of them. This means

your conclusions are in contrast to your discussion in lines 405-415. Overall, I have a concern that you are over-stating your results relative to teh evidence supporting them. **Authors agree that the word misperception was not appropriate, and results are overstated in terms of the measurement period we have and the general body of research on eucalyptus and pine research. We therefore have a contrast between discussion and conclusion. Therefore, the abstract and conclusion has been revised (line 32 to 34, line 416 to 425) to try and fit our results into the broader picture research picture where other studies show eucalyptus to use more than pine over the long-term.**

- There are several cases of sloppy writing. For example, "doesn't" instead of "does not" etc. **We have revised and checked all the language of the paper.**

---

## Author Response (AR4)

Authors would like to thank the editor and reviewers for their contributions to this paper, which have significantly added to the content of the paper.

- Please now sort out the title and resubmit. I think you mean it should be "Transpiration rates from (a) mature Eucalyptus grandis x E. nitens clonal hybrid and Pinus elliottii plantations near the Two Streams Research Catchment, South Africa. **Thank you for the correction and we agree. Authors realized that the word "a" did not match the word "plantations" in the title, therefore we removed the word "a" and kept the word plantations as a plural. The title now reads "Transpiration rates from mature *Eucalyptus grandis* x *E. nitens* clonal hybrid and *Pinus elliottii* plantations near the Two Streams Research Catchment, South Africa".**